# Research Challenges, Recent Advances, and Popular Datasets in Deep Learning-Based Underwater Marine Object Detection: A Review

**DOI:** 10.3390/s23041990

**Published:** 2023-02-10

**Authors:** Meng Joo Er, Jie Chen, Yani Zhang, Wenxiao Gao

**Affiliations:** Institute of Artificial Intelligence and Marine Robotics, College of Marine Electrical Engineering, Dalian Maritime University, Dalian 116026, China

**Keywords:** underwater marine object detection, vision, image quality degradation, small object detection, poor generalization, popular datasets

## Abstract

Underwater marine object detection, as one of the most fundamental techniques in the community of marine science and engineering, has been shown to exhibit tremendous potential for exploring the oceans in recent years. It has been widely applied in practical applications, such as monitoring of underwater ecosystems, exploration of natural resources, management of commercial fisheries, etc. However, due to complexity of the underwater environment, characteristics of marine objects, and limitations imposed by exploration equipment, detection performance in terms of speed, accuracy, and robustness can be dramatically degraded when conventional approaches are used. Deep learning has been found to have significant impact on a variety of applications, including marine engineering. In this context, we offer a review of deep learning-based underwater marine object detection techniques. Underwater object detection can be performed by different sensors, such as acoustic sonar or optical cameras. In this paper, we focus on vision-based object detection due to several significant advantages. To facilitate a thorough understanding of this subject, we organize research challenges of vision-based underwater object detection into four categories: image quality degradation, small object detection, poor generalization, and real-time detection. We review recent advances in underwater marine object detection and highlight advantages and disadvantages of existing solutions for each challenge. In addition, we provide a detailed critical examination of the most extensively used datasets. In addition, we present comparative studies with previous reviews, notably those approaches that leverage artificial intelligence, as well as future trends related to this hot topic.

## 1. Introduction

With constant exploration and usage, natural resources on land have been gradually depleted, driving the hunt for new alternatives. The oceans, which cover about 70% of the planet, have become the next destinations for exploration, as they are magnificent treasure troves of precious resources, providing humans with food, medicine, minerals, and other necessities [1,2,3].

In recent years, the development of marine robots has opened up new opportunities for ocean exploration. When combined with advanced machine vision techniques [4], marine robots have been demonstrated to have significant potential for exploring the underwater environment. In ocean exploration, object detection plays an important role; it is capable of detecting instances of visual objects in digital images, which provide essential information for many downstream tasks. The fundamental question that object detection resolves is: *What objects are where?* [5]. The goal of underwater object detection, as shown in Figure 1, is to predict a bounding box (position) and category label for each object of interest.

Equipped with underwater object detection, marine robots have been widely applied in many real-world applications. For example, in monitoring of marine ecosystems, information about species, size, population density, health state, and other characteristics of marine organisms can be gathered by appropriate underwater object detection techniques, which is significant for decision-making [7]. In management of commercial fisheries, it can be applied to extract important information for cultivation, status surveillance, and early warning of diseases [8]. Underwater object detection is an essential technique for robot grabbing tasks, e.g., picking holothurians, echinus, scallops, and other marine products [9]. Furthermore, underwater object detection plays an important role in the operation of self-driving marine robots, supporting the activities of path planning, collision avoidance and control, etc. These pieces of evidence clearly show that underwater object detection plays a critical role in exploration of the ocean.

In recent years, deep learning techniques, which are capable of learning feature representation from data directly, have attracted great attention in many fields, including underwater object detection. However, due to interference from the underwater environment, such as the complex background structures and characteristics of marine objects and the limitations imposed by exploration equipment, underwater object detection is an extremely challenging task. For instance, the attenuation and scattering effects that occur when light passes through the water can result in substantial degradation in the raw pictures collected by marine robots. On the other hand, most underwater objects, especially marine organisms, are usually very small and tend to congregate in dense distributions, making underwater object detection more difficult. Low-resolution underwater images due to limitations imposed by exploration equipment can result in a loss of information about underwater marine objects. These issues impose great challenges when working with conventional underwater object detection techniques.

In this context, we offer a review of deep learning-based underwater marine object detection techniques. To facilitate a thorough understanding of this subject, we first conduct a rigorous analysis of the research challenges involved in detecting marine objects in the underwater environment. Next, we survey detection techniques that are based on deep learning and present a comprehensive taxonomy within the framework of identified research challenges. In addition, we present a comprehensive discussion of popular datasets and provide a prediction of futuristic trends in underwater marine object detection. Through this paper, readers will be able to arrive at a good grasp of the state of the art in underwater object detection and to appreciate the pros and cons of existing techniques when conducting their own research and development work.

The remainder of this paper is organized as follows. Section 2 discusses related preliminary issues and research challenges. In Section 3, typical deep learning-based object detection techniques in the underwater environment are systematically reviewed within the framework of existing research challenges. Popular datasets and a critical analysis thereof are presented in Section 4. A comparison with previous reviews is provided in Section 5, along with a discussion of future directions in underwater marine object detection. Finally, our conclusions are drawn in Section 6.

## 2. Preliminaries and Research Challenges

### 2.1. Preliminaries

Two kinds of sensors, namely, sonar and cameras, are commonly used in marine exploration. As is well known, it is often more convenient to perform object detection by relying on sound reflections rather than optical information in underwater scenarios. Both sonars and cameras have distinct advantages and disadvantages. Sonar is an acoustic-based exploration device with a range of hundreds of meters [10]. However, only a little information can be carried by images captured by sonar. As illustrated in Figure 2a, it is only possible to identify a vague sketch of the shipwreck shown in the image. On the other hand, optical images captured by a camera contain a great deal of semantic information, which is beneficial for recognition of target objects. The echinus and starfish in the image can be easily recognized from the texture information, as shown in Figure 2b. However, the exploration range of cameras is severely limited by interference in the underwater environment. Another major benefit of optical cameras is their low cost, which has led to their dominance over other sensors.

Therefore, in this review we focus on vision-based underwater marine object detection. In fact, in deep learning, object detection based on sonar images and optical images share the same technology stack except for certain specialized preprocessing methods. We do not dwell on the distinction between sonar images and optical images in this review. On the other hand, marine organisms, shipwrecks, other man-made things, and so forth are among the most interesting objects in the underwater object detection task. In this work, we primarily concentrate on the detection of marine organisms because of their enormous economic worth. Next, we formally define underwater marine object detection and the accompanying evaluation matrices.

#### 2.1.1. Definition of Underwater Marine Object Detection

In underwater object detection, it is necessary to recognize all the objects of interest as well as to locate their positions in the image. As illustrated in Figure 1, position information is generally represented by a rectangular bounding box defined by $\left(x_{i},y_{i},w_{i},h_{i}\right)$, where $\left(x_{i},y_{i}\right)$ denotes the center coordinates of the *i*th object, which measures the top left corner of image as 0-indexed, and $\left(w_{i},h_{i}\right)$ denotes the width and height of the box. Formally, the underwater object detection problem can be formulated as follows:(1)$$X\overset{f(\theta )}{⟶}\left\{\left(p_{i},c_{i},x_{i},y_{i},w_{i},h_{i}\right)\;|\;i\in \left(1,...,N\right)\right\}$$
where f(θ) indicates an object detector that is based on some neural networks parameterized by θ. It takes as its input an image *X*, and finally outputs *N* predictions for objects in that image. Each prediction contains a confidence indicator $p_{i}$, the category label $c_{i}$ that the object belongs to, and the position information encoded in the bounding box, i.e., $\left(x_{i},y_{i},w_{i},h_{i}\right)$. Underwater object detection can provide valuable information for semantic understanding of the underwater environment, and as indicated, it is fundamentally important in the community of marine science and engineering.

#### 2.1.2. Evaluation Metrics

Most of the underwater object detection algorithms are inherited from the community of generic object detection, as are the evaluation metrics. Precision and recall rate are the most widely used evaluation metrics, and are computed based on a confusion matrix, as follows:(2)$$\begin{array}{c} \begin{array}{cc} Precision & =\frac{TP}{TP+FP}\\ Recall & =\frac{TP}{TP+FN} \end{array} \end{array}$$
where TP, FP, and FN denote ’True Positive’, ’False Positive’, and ’False Negative’, respectively, and are defined by the Intersection over Union (IoU) between the prediction bounding box and the ground truth. Different thresholds of IoU result in different precision and recall rates.

In the MS COCO dataset [12], three kinds of thresholds are employed. Precision as averaged across all 10IoU thresholds (IoU=0.50:0.05:0.95) and all categories of objects is called AP or mAP. Accordingly, we have $AP_{50}$ and $AP_{75}$ when the thresholds are set to IoU=0.50 and IoU=0.75. General speaking, prediction can be evaluated across different scales of objects, such as $AP_{S}\;\left(area<32^{2}\right)$, $AP_{M}\;\left(32^{2}<area<96^{2}\right)$, and $AP_{L}\;\left(96^{2}<area\right)$. Recall rates are similarly defined.

### 2.2. Research Challenges

Conventional object detection methods often fail to provide accurate results due to a number of research challenges faced in the underwater environment [13]. To facilitate a comprehensive understanding of this issue, we organize the research challenges into four main categories.

#### 2.2.1. Image Quality Degradation

Raw underwater images captured in the underwater environment usually suffer from the effect of quality degradation, which is mainly caused by selective absorption and scattering of light in a water body [14].

In the transmission of light in water, red light has a longer wavelength than blue and green light, and as such has faster attenuation. This is a phenomenon known as selective absorption, making most underwater images have a bluish or aqua tone, as illustrated in Figure 3. Because the color is distorted, this phenomenon is referred the color distortion problem, and is well known in underwater imagery analysis.

On the other hand, sediments and particles in water can cause a strong scattering phenomenon, which results in blurry and low-contrast images, as illustrated in Figure 4. To highlight this phenomenon, the image can be converted to grayscale to eliminate other sources of interference. However, even in the absence of color distortion, underwater images suffer from strong haziness. Color distortion and blurring lead to serious quality degradation in underwater images.

#### 2.2.2. Small Object Detection

In underwater marine object detection, most of the objects of interest, such as fish schools [15] and benthic organisms [16], are usually very small, and tend to congregate in dense distributions due to their natural habits, as illustrated in Figure 5. This fact leads to the canonical “small object detection trap”, whereby objects only occupy a tiny fraction of an image. In the DUO dataset [6], the vast majority of objects occupy only 0.3% to 1.5% of the image area. Most underwater object detection datasets contain a massive number of small instances.

Small object detection has been a long-standing and challenging problem [18,19]. The low resolution of small objects provides little visual information, and it is difficult to extract discriminative features for locating and recognizing these objects. In addition, there is a significant imbalance between positive and negative samples in small object detection, which is not conducive to model training, as because background samples dominate training loss and the gradient updates are too deviated to learn the features of foreground positive samples.

Meanwhile, it should be highlighted that existing neural network architectures and datasets are not optimized specifically for small object detection. In the hierarchical features of Convolutional Neural Networks (CNNs), increases in convolution and downsampling operations lead to semantic information being enhanced while details are lost, which is very unfavorable for small object detection. Furthermore, there is a lack of large-scale datasets for small objects, making this problem even more challenging.

#### 2.2.3. Poor Generalization

Unlike scenarios on land, there is a huge discrepancy in environmental conditions between different oceans. Images from the natural seabed at Zhangzidao and Jinshitan, Dalian, China are collected in [20], and the collected images are visualized in Lab Color Space [21]. Experimental result shows that there is a huge discrepancy in image distribution between these two ocean areas.

It is generally accepted that most algorithms based on deep learning strongly rely on the i.i.d. assumption between source and target domains [22]. The performance of algorithms often degrades when this assumption is not fulfilled (sometimes known as domain shift), leading to poor generalization. Hence, generic object detectors suffer from seriously degraded detection accuracy when a detector trained for one ocean region is applied to another. The i.i.d. assumption is often violated in practical underwater object detection, creating a major generalization challenge.

#### 2.2.4. Real-Time Detection

Real-time detection is a challenge imposed by the limitations of marine robots. Constrained by the limitations of existing technology, marine robots are usually equipped with embedded platforms that only provide extremely limited computing power. However, deep learning models often require high-performance computing hardware that is not easy to deploy. A standard ResNeXt-50 network has about $25.0\times 10^{6}$ parameters and $4.2\times 10^{9}$ FLOPS on eight of NVIDIA M40 GPUs [23]. The consumption of computing resources has become a crucial challenge when applying deep learning-based detection models to an underwater scenario. We refer to this research challenge as "real-time detection", as it measures whether a deep learning algorithm can meet the requirements of resource consumption, which is the most important prerequisite for practical applications.

#### 2.2.5. Other Challenges

Another challenge in underwater object detection is the inter-class similarity between different species or background due to similar appearance and camouflage types. This inter-class similarity makes it highly infeasible to distinguish these objects, and has become a huge headache for researchers. This problem has been noted by the community of computer vision researchers [24]; however, due to its extreme intractability the problem of inter-class similarity in underwater object detection does not allow for any clear solution. Here, we treat this challenge as a problem for future work, and do not discuss it further.

## 3. Underwater Object Detection Based on Deep Learning

Having presented a systematic discussion of the existing research challenges in underwater marine object detection in the previous section, we now move to a review of the related literature on underwater object detection within the framework of the research challenges identified above. As illustrated in Figure 6, image enhancement is first introduced to deal with the image quality degradation problem, then four techniques are proposed to support small object detection in underwater environments. Next, two different strategies that aim to solve the poor generalization problem are systematically analyzed.

### 3.1. Object Detection Coupled with Image Enhancement

Image quality degradation is the main challenge that differentiates underwater object detection from generic object detection on land. To cope with this problem, image enhancement and restoration techniques have been proposed with the aim of correcting color, improving clarity, and addressing blurring and background scattering [25]. In the literature, image enhancement and restoration are used either as a preprocessing module in the underwater object detection pipeline or are integrated into detection networks.

#### 3.1.1. Image Enhancement as Preprocessing Module

In [26], a typical pipeline in which image enhancement works as a preprocessing module in the underwater object detection pipeline is presented for gripping marine products. As illustrated in Figure 7, an effective image enhancement approach based on the Retinex theory is proposed to enhance the images captured by forward and downward views of marine robots. Next, a real-time lightweight object detector based on SSD [27] is proposed for marine object detection. The locations and categories of objects are determined by the detection result. Through enhanced images, underwater robots can examine the seabed environment more precisely.

The image enhancement method presented in this pipeline is based on the improved multi-scale Retinex method with color preservation (IMSRCP) [28]. In IMSRCP, color pre-correction is first used to balance the color of damaged underwater photos and decrease dominant color. Next, to estimate the reflectance and illumination component, the enhanced multi-scale Retinex is employed in conjunction with intensity computation via channels. Furthermore, the picture is recovered from the logarithmic domain while the compensation dynamics are adjusted. Finally, the color of the original image might be kept depending on the actual application.

Such a pipeline that enhances images before object detection has been a dominant approach in the underwater object detection community for a long time, and research works in this area are ongoing. In [29], the combination of a Max-RGB filter [30] and the Shades of Gray method [31] for improving underwater vision was used to enhance underwater images. Next, a CNN was proposed to handle the weak illumination problem in underwater images. Following image processing, a deep CNN detector was proposed for underwater object detection and classification. Experimental results demonstrated that the improved underwater vision system can assisting a robot in executing underwater missions more efficiently. Similar works on underwater object detection include [32,33,34].

Underwater image degradation is mainly due to the selective absorption and scattering effect, which can be depicted by a physical image formation model. In [35], a widely used Simplified Model is proposed. As illustrated in Figure 8, natural light penetrates from the air to an underwater scene, and direct transmission reflected off the object can propagate towards the camera. Another portion of the light meets suspended particles, causing much scattering. As such, the radiance perceived by the camera is the sum of two components, that is, the background light formed by multi-scattering and the direct transmission of the reflected light.

It is well known that light with different wavelengths are attenuated at different rates in water. Therefore, the image formation model illustrated in Figure 8 can be formulated as follows:(3)$$\begin{array}{cc} I_{\lambda }\left(x\right) & =J_{\lambda }\left(x\right)\cdot t_{\lambda }\left(x\right)+\left(1-t_{\lambda }\left(x\right)\right)\cdot B_{\lambda },\lambda \in \left\{r,g,b\right\},\\ t_{\lambda }\left(x\right) & =exp(-\beta_{\lambda }d\left(x\right)) \end{array}$$
where *x* denotes a point in the scene, λ denotes the red/green/blue component of the light, $J_{\lambda }\left(x\right)$ is the scene radiance from *x*, $t_{\lambda }\left(x\right)$ is the transmission of $J_{\lambda }\left(x\right)$ after the light is reflected and has reached the camera, $B_{\lambda }$ is the background light, and $I_{\lambda }\left(x\right)$ denotes the captured image. Here, $t_{\lambda }\left(x\right)$ depends on the object–camera distance d(x) and the water attenuation coefficient of each channel $\beta_{\lambda }$.

Most common scenes of quality degradation in underwater images can be explained based on the image formation model. The different attenuated rates of light describe the selective absorption, which leads to the problem of color distortion. The scattering phenomenon is depicted by the background light component $B_{\lambda }$, which can provide insight for solving this problem.

In [36], Dana et al. proposed an advanced underwater single-image color restoration method based on a physical image formation model. As illustrated in Figure 9, veiling light pixels are first detected by structured edges and used to calculate the background light component *B* for de-scattering. The object–camera distance is measured by stereo images. Unlike existing works, Dana et al. considered different Jerlov water types in order to estimate transmissions based on a haze-lines model. Finally, color corrections were performed using a physical image formation model and the best result was selected from different water types.

There are other complicated image formation models, such as the revised model proposed in [37] that can depict the process of real-world underwater imaging more precisely by considering the impacts of sensors, ambient illumination, etc. However, due to its complexity, the revised model has received barely any attention, and the simplified model continues to play the vital role in underwater imaging research.

#### 3.1.2. Image Enhancement Integrated into Object Detection Networks

Image enhancement has proven to be helpful for conventional hand-crafted features (e.g., HOG [38], SIFT [39]). How visual enhancement has contributed to deep-learning-based object detection in underwater scenes is discussed in [20]. The authors of [20] showed that enhancement cannot improve within-domain detection accuracy due to the degrading recall efficiency; however, it provides a basic guarantee of generalization in cross-domain scenarios. Many researchers believe that this is caused by the inconsistency resulting from decoupled image enhancement and object detection. As a consequence, another solution for the image quality degradation problem is to integrate image enhancement and object detection into a single model via multi-task learning [40].

In [41], a lightweight deep neural network was proposed for joint learning of color conversion and object detection from underwater images. As illustrated in Figure 10, to alleviate the problem of color distortion, an image color conversion module is first employed to convert color pictures into grayscale images. Then, object detection is performed on the converted images. Color conversion and object detection are learned jointly by a combined loss function:(4)$$\begin{array}{c} L_{Total}=\lambda_{TV}\times L_{TV}+\lambda_{feature}\times L_{feature}+\lambda_{style}\times L_{style}+\lambda_{detection}\times L_{detection} \end{array}$$

The color-converted image is measured by three different loss function, namely, the total variation (TV) loss $L_{TV}$ [42], feature reconstruction loss $L_{feature}$ [43], and style reconstruction loss $L_{style}$ [43]. To train the multi-scale object detector, the loss function in YOLO v3 [44] is employed directly, denoted as $L_{detection}$, while $\lambda_{TV}$, $\lambda_{feature}$, $\lambda_{style}$, and $\lambda_{detection}$ respectively denote the four weight coefficients. With the combined loss function, color conversion is guided in a direction that boosts the performance of object detection.

A comparison is summarized in Table 1. Here, “GFLOPs” denotes gigaflops (floating point operations per second), which reflects the computation complexity of the model, and “Epochs” denotes the number of training epochs. As illustrated in Table 1, experimental results show that the proposed deep model outperforms models without color conversion or with standard color conversion as a preprocessing module when tested on the authors’ custom dataset for object detection. Without interference from color distortion, object detection performance can be improved while maintaining low computational complexity.

Another similar work is [45], where an end-to-end marine organism detection framework was developed to handle underwater image quality degradation problems caused by noise pollution, color cast, and motion blurring. As illustrated in Figure 11, the framework consists of underwater image enhancement, feature extraction, and back-end detection. Three submodules are employed in the underwater image enhancement module to complete step-by-step denoising, color correction, and deblurring enhancement. Notably, three submodules must be jointly pretrained and optimized with object detection. With image enhancement, the capability of the detector to deal with highly damaged underwater images is significantly promoted. Experimental results show that the proposed framework can improve detection precision by at least 6% compared to existing models for marine organism detection without image enhancement.

Unlike integration of image enhancement using the combined loss function, the Composited FishNet proposed in [46] provides a novel and unified solution for alleviating interference in the underwater environment caused by variance in picture brightness, fish orientation, seabed structure, aquatic plant movement, fish species form, and texture differences.

In Composited FishNet, the authors propose a new composite backbone network, illustrated in Figure 12. The scene change information is encoded by an auxiliary backbone network from the background images without fish; next, the learned background feature is subtracted from the of upper layer feature in the main backbone network. Experimental results show superior performance when the interference is eliminated. Compared with integration through the combined loss function mentioned above, the strategy in Composited FishNet that models the image enhancement process in the neural network is both more elegant and more powerful.

#### 3.1.3. Summary

In this section, we have reviewed two different strategies for solving image quality degradation in underwater object detection. The most prominent characteristic of image enhancement using a preprocessing module is that image enhancement and object detection are decoupled. This decoupling sometimes leads to unexpected results, such that even images with high perceptual quality may not be able to achieve excellent performance in object detection. One possible explanation is that decoupled image enhancement is not desirable for the object detection task. Thus, the second strategy involves joint learning of image enhancement and object detection, which is a more promising technique. In this paradigm, joint learning guides the enhancement module in a direction that can boost detection performance. Research on this topic is in its infant stages, and additional efforts should be directed towards research into joint learning.

### 3.2. Detection of Small Objects

Small object detection has always been a priority in the marine community due to the characteristics of underwater marine objects, as illustrated in Section 2. Here, we outline the four pillars of small object detection in underwater environments, namely, multi-scale representation, contextual information, super-resolution, and balance of positive and negative examples.

#### 3.2.1. Multi-Scale Representation

In the hierarchical features of CNNs, low-level detailed features are useful for object localization, while semantic information in high-level feature maps is significant for classification of objects. Obviously, most features of a small objects are lost in high-level feature maps, making small object detection extremely difficult. In the literature, a number of researchers have proposed multi-scale feature fusion to generate a more discriminative representation for small object detection.

The first canonical solution for multi-scale representation is the feature pyramid network (FPN) [47]. As illustrated in Figure 13, high-level feature maps are upsampled and added to lower-level ones in order to generate an information-rich representation that is significant for detecting and classifying small objects.

The FPN has long been a fundamental block in most deep-learning architectures for underwater object detection. In [9], an efficient detector (termed AquaNet) based on CenterNet [48] was presented, in which multi-scale feature maps from different stages are fused by the FPN to recognize masses of tiny objects from foggy underwater images. In [49], a Multi-scale ResNet (M-ResNet) model based on a modified residual neural network for underwater object detection was proposed. The M-ResNet leverages multi-scale feature fusion by the FPN to allow accurate detection of objects of different sizes, especially small ones. Experimental results show that the proposed approaches always provide improved performance thanks to enhancement of predictive feature representations created by the FPN.

The traditional FPN, on the other hand, is constrained by its one-way top-down information flow. A bottom-up path augmentation can be applied to enrich the entire feature hierarchy in the Path Aggregation Network (PANet) [50] with precise localization signals in lower layers by shortening the information path between the lower layers and the topmost feature. In [50], PANet was integrated with FPN, complementing it in a way that allowed a mixed multi-scale feature structure to be generated, which was then used in an uneaten feed pellet detection model for aquaculture. Experimental results show that the mAP increased by 27.21% compared to the original YOLO-v4 method [51].

To solve the problem of insufficient utilization of semantic information caused by linear upsampling in PANet, an Enhanced Path Aggregation Network (EPANet) was proposed in Composited FishNet [46] for automatic detection and identification of fish from underwater videos. Compared with PANet, the main differences are twofold: (1) a jump connection method is used to merge the backbone network output with the EPANet output information, and (2) the linear interpolation upsampling method is replaced by PixelShuffle [52] to avoid information loss.

Table 2 illustrates a comparison of different feature fusion approaches for underwater object detection on the SeaLife 2017 dataset [46]. Experimental results show that the EPANet exhibits a 1.4% increase in mAP over FPN under the same condition. Compared with PANet, the accuracy is increased by 0.8% due to nonlinear fusion. EPANet outperforms alternatives in terms of the $AP_{50}$ metric, though it suffers from slight degradation in the $AP_{75}$ metric. Comprehensive experiments have demonstrated the effectiveness and efficiency of the EPANet module.

Unlike EPANet, the Bidirectional Feature Pyramid Network (BiFPN) [53] is enhanced by removing nodes with only one input edge. These nodes contribute less to the representation that seeks to fuse various features. As illustrated in Figure 14, white nodes represent feature maps from backbone network, arrows represent information flow, and colored nodes represent fused representation. Meanwhile, the BiFPN layers are repeated multiple times to allow additional high-level feature fusion. It is worth mentioning that weighted feature fusion is used in the BiFPN, as feature maps of different resolutions usually contribute to the output representation unequally.

Following the improvements in BiFPN, Faster R-CNN was adapted in [54] for underwater organism detection. The authors created a ResNet–BiFPN structure to enhance the capability of feature extraction and multi-scale feature fusion. On the URPC dataset [55], the detection accuracy was 8.26% higher than the original Faster R-CNN. Experimental results showed that BiFPN was able to achieve better accuracy with fewer parameters and FLOPs.

On the other hand, inspired by shortcut connections in residual neural networks [56], a Shortcut Feature Pyramid Network (S-FPN) was proposed in [57] to improve an existing multi-scale feature fusion strategy for holothurian detection. A shortcut connection was attached to the FPN model, with auxiliary information flow merged into conventional multi-scale feature fusion. Shortcut connection enhances feature fusion and reduces information loss in deep networks. Experimental results showed that the mAP of S-FPN reaches 91.5%, outperforming baseline methods on the authors’ custom datasets.

Unlike conventional feature fusion, which crosses multiple stages of the backbone, the idea of fusing multi-scale features that cross different channels using different sizes of kernels or complex connections in one block was proposed in [58,59]. This is a method for enhancing the representational learning ability of a neural network by using the idea of multi-scale feature fusion rather than feature fusion across multiple stages of the backbone network.

A Multi-scale Contextual Features Fusion (MFF) block was proposed in AquaNet [9]. In the MFF module, as illustrated in Figure 15, the input is first expanded by 1×1 convolution, then split into groups by channel allocation. Each group is convoluted by a kernel, with different kernel sizes used for extraction of multi-scale features. Shortcut connections between different grounds are employed for feature enhancement. For final feature fusion, outputs from all grounds are concatenated and convoluted by 1×1 kernels.

Most of the aforementioned FPN-based modules and their variants always suffer from a deficiency of context-independent characteristics. There is a substantial risk of information redundancy and negative influence in such feature representations for specific tasks. To alleviate this problem, the attentional mechanism can be integrated into the feature fusion module.

In [60], a Multi-scale Attentional Feature Fusion Module (AFFM) was proposed to fuse semantic and scale-inconsistent features between different convolution layers. As illustrated in Figure 16, feature maps from different stages of the backbone network are first summed and sent into the Multi-Scale Channel Attention Module (MS-CAM). The MS-CAM outputs the attention scores through a Sigmoid function. Finally, attention scores are used as references for weighted aggregation of multi-scale features. Experimental results showed that attentional multi-scale feature fusion is superior in underwater object detection.

Similarly, in [61], J. Chen et al. claimed that multi-scale representation should be both spatially aware and scale-aware, and presented a dynamic feature fusion module based on spatial attention and scale attention. This method can fuse different scale feature maps adaptively, thereby enhancing the ability to detect small marine objects by a considerable amount.

From the aforementioned discussion, it is apparent that multi-scale representation for small object detection in underwater environments has been well-studied. However, as an efficient technique of information aggregation, attentional mechanisms have not been studied enough. This creates a need for further investigation of attention-based multi-scale feature fusion. Fully exploiting the potential of attention is expected to effectively promote the development of underwater object detection.

#### 3.2.2. Contextual Information

It is widely acknowledged that lack of visual information severely degrades the performance of small object detection. Many researchers believe contextual information can aid in detecting small objects by leveraging the relationship between objects and their coexisting environment.

(1)Spatial Contextual Information

In the paradigm of anchor-based object detection, the most natural way of extracting contextual information is to enlarge the candidate boxes, which means taking more surrounding environmental information into consideration. In [62], contextual information was extracted from a context region 1.5 times larger than the proposed candidate region. Similarly, in [63], four region with ‘foveal’ fields of view of 1×, 1.5×, 2× and 4× of the original proposed box that are all centered on the object of interest were added to the proposed model in order to incorporate contextual information. Experimental results showed that the introduction of spatial contextual information can always result in improved performance.

However, what is noteworthy is that incorporating contextual information for small object detection does not mean taking into account the entire background region, which is redundant and ambiguous, without any restraints. On the contrary, extracting predictive features more precisely is crucial for small object detection.

Deformable convolutional networks are often used to detect marine organisms that are in variable scale forms (e.g., echinus, starfish). In [64], it was employed to improve the capability of the SSD detector in detecting fish, bionics, and other objects in the underwater environment. With precise feature extraction, the detection accuracy and speed in complex underwater environments can be improved significantly.

In [65], two drawbacks that degrade performance of the single-stage detector were identified: first, the detection head regresses coordinates from pre-defined anchors directly, although most anchors are far from matching object regions; second, the information used for classification likely originates from erroneous places, where characteristics may not be precise enough to describe objects. To alleviate these two problems, a novel neural network based on anchor refinement and feature location refinement mechanism was proposed in [65].

As the left image in Figure 17 shows, classic SSD-style detectors rely on hand-crafted anchors that are stiff and invariably inaccurate. The Anchor Refinement Module (ARM) generates refined anchors that provide better initialization for the second-step regression. A location head performs convolution to generate an anchor offset ar using backbone-based ARM features $f_{ARM}$, i.e.,
(5)$$ar=W_{ar}*f_{ARM},$$
where the anchor offset ar is the coordinate offset from original anchors, $W_{ar}$ is the learnable convolutional weight, and ∗ denotes convolution. Anchor refinement is urgently needed, because on one hand it greatly relieves the difficulty of localization, while on the other it can guide the refinement of feature location.

The original SSD employs a normal 3×3 grid *R* (kernel) to estimate the category probability and the coordinates of a feature map cell. Thus, the prediction is provided by
(6)$$P_{p_{0}}=\sum_{p\in R}w\left(p\right)\cdot f_{ODM}\left(p\right),$$
where *P* denotes the prediction of category probability or coordinate offset, *w* denotes the convolution weight, $f_{ODM}$ represents features from object detection module, *p* indicates positions in *R*, and $p_{0}$ is the center of the respective field.

It is believed that the respective field of *R* usually fails to describe the refined anchor region. As such, allowing *R* to deform to fit various anchor changes, the deformable convolution is developed to capture accurate features with the feature offset δp, as follows:(7)$$P_{p_{0}}=\sum_{p\in R}w\left(p\right)\cdot f_{ODM}\left(p+\delta p\right)$$
where the feature offsets $\Delta p=\left\{\delta p\right\}$ are predicted based on anchor offsets in the Anchor Refinement Module, i.e., feature location refinement is provided by
(8)$$\Delta p=W_{fr}*ar$$
where $W_{fr}$ denotes another learnable convolutional weight and ar is the anchor offset from the ARM module.

Based on the anchor refinement and feature location refinement mechanisms, a Dual Refinement Network (DRNet) has been developed for high-performance detection and applied to online underwater object detection and grasping with an autonomous robotic system.

Spatial context information describes the relationship between objects and their surrounding environment. By leveraging this relationship, objects can usually be recognized more easily, as their camouflage is usually not perfect enough to blend in with the environment. However, the extraction of context information must be carried out cautiously and meticulously, as inappropriate operations are noisy for the detection of underwater marine objects.

(2)Temporal Contextual Information

In addition to spatial contextual information, features from previous frames of underwater video are useful for detecting small objects in the current frame. These features are referred to as temporal context information.

Motion information is an important piece of temporal context information. In [66], motion information was obtained by Gaussian Mixture Modeling (GMM) [67] and optical flow [68] from adjacent frames. As illustrated in Figure 18, by leveraging temporal context information in combination with the original image, a more accurate fish detector based on Faster R-CNN was built for underwater object detection.

The GMM model represents a probability density function $P(x_{t})$ of data *x* at time *t* as a weighted sum of multiple individual normal distributions $g(x_{i})$ for pixel *i*. The background features can be learned by GMM in a statistical model using the mean and covariance values of the pixel data. Any random and rapid changes in the pixel values of a frame according to the GMM produces a mismatch with the background model of that pixel, in which case motion is presumed to be detected.

The optical flow is represented by a 2D motion vector in the video footage caused by 3D motion of the moving objects. Optical flow is captured by Taylor series approximation with partial derivatives to spatial and temporal coordinates [69]. Based on the optical flow, the proposed method captures a motion between two successive video frames at times *t* and t+δt at every position.

In [70], unlike traditional techniques such as GMM and optical flow, temporal context information is learned by a deep neural network automatically. As illustrated in Figure 19, in order to leverage rich temporal information in videos, each frame of the video images is first processed by a CNN for feature extraction, then these features are fused using the LH-LSTM module. Through the Conv-LSTM model, temporal contextual information is learned by a neural network automatically and integrated from previous frames for current detection. The ConvLSTM model provides a real-world object detection maneuver for underwater object gripping, providing significant proof of the effectiveness and efficiency of using temporal context information.

In addition to low-level vision features, detection results for the same object in previous frames can provide stronger confidence for detection in future frames. In [71], to preserve rare and endangered objects or eliminate exotic invasive species, a method based on the YOLO framework [72] was proposed for properly classifying objects and counting their numbers in consecutive underwater video images. This approach accumulates object classification results from previous frames to the present frame. The cumulative mean for each object $Avg_{i}\left(k\right)$ can be computed as follows:(9)$$Avg_{i}\left(k\right)=\frac{\left(i-1\right)}{i}Avg_{i-1}\left(k\right)+\frac{p_{i}\left(k\right)}{i}$$
where *i* denotes the number of frames, *k* denotes a classification object, $p_{i}\left(k\right)$ is the probability of classification for the *k*th object on the *i*th frame, and $Avg_{i}\left(k\right)$ denotes the cumulative mean. By applying this heuristic method to YOLO, the proposed method can obtain higher accuracy than the original YOLO detector, which uses only one frame of video images for object detection.

From the aforementioned discussion, it is apparent that both low-level vision information and detection results have been demonstrated to provide effective temporal context information. This is conducive to detection of small underwater objects. However, extraction of temporal context information from previous video frames requires additional computation, resulting in higher computational complexity, which is undesirable in marine robotics.

#### 3.2.3. Super-Resolution

It was demonstrated in [73] that fine details of a small objects for localization can be recovered by Generative Adversarial Networks (GANs). This leads to our discussion of the third pillar of small object detection, i.e., super-resolution, which revolves around converting a low-resolution image with coarse details into a high-resolution image with better visual quality and refined details. In this way, the lack of visual information for small object detection can be circumvented.

In the literature, many deep-learning-based techniques have been employed for super-resolution, including linear networks, residual networks, recursive networks, and more [73]. Linear networks map low-resolution (LR) images to high-resolution (HR) images using a single path for signal flow. The differences between LR and HR images are learned by convolution layers [74]. Super-Resolution Convolutional Neural Network (SRCNN) is one of the outstanding works on super-resolution techniques that rely on pure convolutional layers [75].

In [76], SRCNN is employed to obtain good-quality underwater images in low-light conditions. As illustrated in Figure 20, to obtain LR components, raw data was iteratively processed by Total Variation (TV) regularization [77]. Next, bicubic interpolation, a preprocessing technique for CNNs, was used to increase the LR components. Thereafter, three different convolutional layers were employed to extract features and reconstruct HR patches. All HR patches were merged to generate suitably high HR components, and HR feature vectors were rebuilt. Lastly, the rebuilt pictures were subjected to post-processing, such as noise filtering. Experimental results showed that image reconstruction by the SRCNN method can increase the number of total pixels by about 30%.

Residual networks attach skip connections to the network to learn residues, i.e., high-frequency components between the input and ground truth. Several networks have provided a boost to super-resolution performance using residual learning [78,79]. In [80], a residual-in-residual network-based generative model (Deep SESR) was suggested to handle simultaneous enhancement and super-resolution for underwater robot vision. This model can be trained to restore perceptual picture quality at 2×, 3×, or 4× higher spatial resolution. Residual connection is extensively employed in the feature extraction network of the Deep SESR model, and deconvolution is performed for reconstruction of high-resolution images. Similarly, a deep generative model based on the residual network for single picture super-resolution of underwater photography was presented in [81] that is suitable for object detection in marine robots. Compared with the linear network, the representation capability of residual networks is enhanced significantly by the residual connection, which is desired by super-resolution.

Many other promising techniques have exhibited their effectiveness in super-resolution, though there has been less exploitation in underwater object detection. Recursive networks break down harder super-resolution problems into a set of simpler ones that are easy to solve [82]. Progressive reconstruction design is a promising choice for super-resolution by predicting the output in multiple steps, i.e., 2× followed by 4× and so forth [83]. Super-resolution based on densely connected networks aims at utilizing hierarchical cues available along with the network depth in order to gain more flexibility and richer feature representation [84]. Multi-branch networks in super-resolution aim to obtain a diverse set of features at multiple context scales. Such complementary information can then be fused to obtain a better HR reconstruction, as in [85]. The attention model can boost super-resolution performance by selectively attending to essential features [86].

By recovering fine detail information, super-resolution can help to improve the performance of small object detection in the underwater environment. Due to space constraints, we can only provide a brief introduction to super-resolution techniques here. For more information, please refer to [14].

#### 3.2.4. Balancing of Positive and Negative Examples

There is always an imbalance between positive and negative examples, and small object detection makes this problem more serious. In small object detection, an increase in candidate locations is generated on images, resulting in an extreme class imbalance [19]. In the class imbalance scenario, the easily categorized negative samples overwhelm the training loss; in this case, the direction of gradient updates becomes deviated, resulting in both feature learning of foreground positive samples and the overall training procedure being more difficult. To alleviate this problem, two crucial strategies, sampling and cost-sensitive loss function, have been taken into consideration.

Sampling strategies, accompanied by label assignment procedures, have been developed for a long time to alleviate the class imbalance problem in small object detection. In the training process of Faster R-CNN, the ratio of positive and negative samples in the stage of region proposal is kept at 1:1 by random sampling, and the ratio in the detection head is kept at 1:3 [87]. However, random sampling generally leads to simple samples dominating selected samples. To address this case, an IoU-balanced sampling strategy was proposed in [88] to solve the class imbalance problem more elegantly.

Assume that we need to sample *N* negative samples from *M* matching candidates. Under random sampling, the selected probability for each sample is as follows:(10)$$p=\frac{N}{M}$$

In the IoU-balanced sampling approach of [88], the sample interval is split into *K* bins according to IoU in order to increase the chosen likelihood of hard negatives, and the *N* required negative samples are spread evenly throughout the bins. Next, samples are uniformly chosen from the bins. The probability of each sample being selected is computed as follows:(11)$$p_{k}=\frac{N}{K}*\frac{1}{M_{k}},\;k\in \left[0,K\right)$$
where $M_{k}$ is the number of sampling candidates in the corresponding interval denoted by *k*. Experimental results show that IoU-balanced sampling can guide the distribution of training samples to be as near as possible to the distribution of hard negatives, which helps to alleviate the class imbalance challenge. Many other heuristic methods have been employed for sampling [62], although we do not address them further here due to space constraints.

An alternative solution to class imbalance is to design a cost-sensitive loss function with which different penalty factors (weights) for different prediction errors can be used to adjust the balance of classes without increasing computational complexity. The most commonly used method for this solution is Focal Loss (FL) [89], which reshapes cross-entropy loss to a counterpart that downweights the loss assigned to well-classified examples. Specifically, FL is defined as follows:(12)$$\mathrm{FL}\left(p_{t}\right)=-\alpha_{t}{\left(1-p_{t}\right)}^{\gamma }\log\left(p_{t}\right)$$
where $p_{t}$ is the estimated probability, α is a weighting factor to balance the importance of positive/negative examples, and γ indicates a focusing parameter that adjusts the rate at which samples are downweighted smoothly, achieving a balance between hard and easy samples. Due to its superior performance, FL has been used as the de facto loss function in the anchor-free paradigm of object detection [90], and is widely used in the underwater environment [91].

Compared with FL, the Piecewise Focal Loss (PFL) function developed in [57] pays more attention to the difference between the loss weights of hard samples and easy samples. It concentrates more on the training difficulty of hard samples. When the ground truth is 1, those examples with predictive confidence smaller than 0.5$\left(p_{i}<0.5\right)$ are referred to as hard samples, as they are hard for the detector to classify. On the contrary, the other samples with predictive confidence greater than 0.5(pi≥0.5) are easy samples. When the ground truth is 0, the interpretations are the other way around. Based on the distinction between hard and easy samples, the PFL is formulated as follows:(13)$$\begin{array}{c} PFL\left(p_{i},p_{i}^{*}\right)=\left\{\begin{array}{cc} -{\left(1-p_{i}\right)}^{\gamma^{\prime }}logp_{i},\; & if\;p_{i}^{*}=1\\ -p_{i}^{\gamma^{\prime }}log\left(1-p_{i}\right),\; & if\;p_{i}^{*}=0 \end{array}\right.\\ where\;if\;p_{i}^{*}=1,\gamma^{\prime }=\left\{\begin{array}{cc} -1/\gamma ,\; & if\;p_{i}<0.5\\ \gamma ,\; & if\;0.5\geq p_{i}\leq 1 \end{array}\right.\\ \;if\;p_{i}^{*}=0,\gamma^{\prime }=\left\{\begin{array}{cc} \gamma ,\; & if\;p_{i}<0.5\\ -1/\gamma ,\; & if\;0.5\geq p_{i}\leq 1 \end{array}\right.\\ p_{i}^{*}=\left\{\begin{array}{cc} 0,\; & negative\;label\\ 1,\; & positive\;label \end{array}\right. \end{array}$$
where $p_{i}$ represents the probability of each candidate box being predicted as an object, $p_{i}^{*}$ represents the label of the ground truth, 1 denotes a positive label, and 0 denotes negative label. The term $\gamma^{\prime }$ is the adjustment factor that is used to adjust exponential weights of different confidence intervals. This strategy makes the network more inclined to learn from hard samples. Experimental results have demonstrated superior performance of PFL, achieving a mAP of 92.3%, outperforming the baseline cross-entropy of 91.5% and FL of 91.8% on a custom dataset.

There are further studies on cost-sensitive loss functions that have paid attention to different topics, such as hard example mining [92], gradient harmonizing mechanism [93], and long-tail imbalance [94]. However, we mainly focus on sample imbalance between positive and negative examples for small object detection. Due to space constraints, further discussions are omitted.

#### 3.2.5. Summary

In this section, the four pillars of small object detection in underwater environment have been systematically reviewed. Multi-scale representation aims to fuse multi-scale features for discriminative prediction. Attentional multi-scale feature fusion is a research area that is expected to be further explored for the promotion of underwater small object detection. Both spatial contextual information and temporal contextual information are conducive to small object detection by leveraging the relationship between objects and their coexisting environment. However, inappropriate operations on contextual information are be noisy, and result in higher computational complexity. By recovering fine detail information, super-resolution improves the performance of small object detection. Complementing super-resolution with image enhancement is a promising direction for underwater small object detection research. The last pillar, proposed for solving the imbalance between positive and negative samples, involves respective sampling strategies and cost-sensitive loss functions. Both of these can guide the detector to focus on learning features from positive samples, thereby avoiding being overwhelmed by the much larger number of negative samples.

### 3.3. Generalization in Underwater Object Detection

Generalization is a crucial challenge in underwater object detection and is highly important for real-world applications, as domain shifts can dramatically degrade detection performance. In general, a large-scale dataset with high diversity can address the generalization problem. However, massive and varied underwater scenes make domain-diverse data collection impossible. In this context, two strategies for improving generalization, namely, data augmentation and domain transformation, are discussed.

#### 3.3.1. Data Augmentation

Data augmentation is a basic yet effective technique for improving generalization. It refers to a set of approaches that are capable of increasing the diversity of training datasets [95], such as MixUp [96], Mosaic [51], and PhotoMetricDistortion [97]. These approaches have been fully explored, and have proven to be effective methods of improving generalization in deep learning [98].

In [99], three different data augmentations were employed to train a Faster R-CNN detector for identification and recognition of marine organisms. First, inverse restoration was adopted to mimic various kinds of marine turbulence. Next, perspective transformation was exploited to simulate diverse camera shooting perspectives. Finally, illumination synthesis was employed to simulate varied and uneven lighting situations in underwater environments. Leveraging these three data augmentations, an improvement in both the generalization and robustness of the resulting Faster R-CNN for underwater marine object detection was confirmed by experimental results.

In [100], a detector for detecting marine organisms based only on a tiny underwater dataset with a limited water quality category was proposed. A data augmentation technique based on Water Quality Transfer (WQT) was developed to expand the domain diversity of the initial dataset. WQT is a wavelet-corrected transfer that is based on whitening and coloring transforms. Leveraging WQT, the URPC dataset was transferred to different types of water quality, and promising generalization in underwater object detection was achieved.

In addition to traditional augmentation techniques, further research based on deep learning was conducted in [101,102]. In [101], an approach called Region of Interest Mixed (RoIMix) was proposed to create different training examples by fusing several pictures at the Regions of Interest (RoI) level. RoIMix seeks to mimic overlapping, occlusion, and blurring, allowing the model to implicitly learn to recognize underwater organisms in different conditions.

Let $x\in R^{H\times W\times C}$ and *y* represent a proposal and its label. RoIMix seeks to combine two random RoIs, $\left(x_{i},y_{i}\right)$ and $\left(x_{j},y_{j}\right)$, which when extracted from a batch of images generate virtual proposals $\left(\tilde{x},\tilde{y}\right)$:(14)$$\begin{array}{cc} \tilde{x} & =\lambda^{\prime }x_{i}+\left(1-\lambda^{\prime }\right)x_{j},\;\tilde{y}=y_{i},\\ \lambda & =B(a,a)\\ \lambda^{{}^{\prime }} & =max(\lambda ,1-\lambda ) \end{array}$$
where B(a,a) is a Beta distribution with parameter *a*, as $x_{i}$ is assigned with a larger coefficient $\lambda^{\prime }$ and $y_{i}$ is used to label virtual proposal $\tilde{x}$.

For example, we have $x_{1}$ and $x_{2}$, which are two RoIs containing a scallop and an echinus, respectively; $x_{1}$ is in a blurring environment, while $x_{3}$ denotes an occluded sample cropped from a training image, e.g., it represents an echinus lying on a scallop. RoIMix combines $x_{1}$ and $x_{2}$ to generate a new virtual proposal $\tilde{x}$, similar to $x_{3}$, which is used to simulate occlusion and blurring. Experimental results demonstrate that the RoIMix can enhance detection performance on the URPC dataset and Pascal VOC dataset, achieving results 1.18% mAP and 0.8% mAP better, respectively, than the original Faster R-CNN model.

The above proposals can be refined to construct a high-quality training sample for implicitly good generalization. In [102], a proposal-refined object detector was proposed to deal with the detection task in underwater images, which is characterized by insufficient annotation. First, a segmentation network with a poor match for foreground–background segmentation was developed. Next, segmentation was utilized to enhance the region proposals and generate a high-quality training dataset. Leveraging this segmentation, proposal refinement can potentially improve the generalization of object detection.

Traditional augmentation techniques usually work as pre-processing procedures in the detection pipeline, and have become an important cornerstone for superior performance and generalization of underwater object detection. Modern augmentations that leverage deep neural networks represent promising techniques for achieving robust generalization.

#### 3.3.2. Domain Transformation

Domain transformation is an opportunistic strategy for archiving good generalization in underwater object detection by transforming images from distinct domains to middle representations or transforming the trained detector from the source domain to a target domain, as illustrated in Figure 21.

In [20], it was shown that distributions of two domains are highly overlapped after the same GAN-based restoration (GAN-RS) is applied to images from different ocean regions. By adopting the GAN-based restoration approach of [21], the problem of domain shift is bypassed subtly. It was shown that detectors developed for one ocean are capable of completing the detection job in another ocean region.

Another experiment using the weak filtering-based restoration approach of [103] was carried out in [20]. Experimental results showed that the generalization problem is alleviated, though the performance does not compare favorably with GAN-based restoration. In [20], it was demonstrated that images from different domains can be transformed into a middle domain representation through enhancement and restoration by the same method; favorable generalization in underwater object detection was achieved. It is apparent that the problem of domain shift can be gradually alleviated by increasing the restoration intensity. As a contribution to domain shift suppression, visual restoration is essential for object detection in underwater environments.

Numerous visual restoration (enhancement) methods have been developed as preprocessing procedures in the automatic pipeline of visually-guided underwater object detection [25]. Visual restoration can provide superior performance for conventional underwater object detection, as richer information can be obtained from enhanced images, and can achieve robust generalization, as it bypasses the problem of domain shift.

Transfer learning is an alternative approach for domain transformation that can achieve good generalization in underwater object detection by storing knowledge gained from the source domain and applying it to a different related target domain. Transfer learning is helpful for handling scenario involving machine learning with a lack of labeled training data.

In [104], transfer learning was employed to alleviate the scarcity of labeled marine data. A marine object detector was pretrained on the ImageNet and MS COCO datasets. In the pre-training technique, the learned knowledge was stored in the parameters of detector and applied to marine object detection, leading to a high-quality detection result with a relatively low number of labeled images. Naturally, transfer learning is a promising technique for addressing the generalization challenge raised by domain shift between ocean regions, as it can transfer the knowledge gained from the source domain to the target domain. It has been proven that transfer learning is extremely effective in underwater object detection [104].

Domain transformation achieves robust generalization in a clever way, helping to achieve significant breakthroughs in underwater marine object detection; as such, further investigation is highly desirable.

#### 3.3.3. Summary

In this section, data augmentation and domain transformation techniques for alleviating the poor generalization challenge raised by domain shift in underwater object detection have been discussed. Both are indispensable for practical applications in real world. In the practical scenario, there is always a lack of labeled data; furthermore, domain shifts occur between the training and test data. Data augmentation can help to increase the diversity of the training data, and modern techniques that leverage deep neural networks have exhibited great potential in such augmentation. Domain transformation helps to bridge the gap in distributions between different domains. Image enhancement and transfer learning are two techniques that can implement domain transformation. Further investigation could help to achieve significant breakthroughs in underwater marine object detection.

### 3.4. Real-Time Detection

Real-time detection is a crucial indicator that determines whether deep learning models can really work in practical applications. Hence, there is a never-ending quest for excellence in real-time performance of underwater object detection. In this context, we briefly review the evolution of deep leaning-based detectors in underwater object detection and summarize two key techniques for real-time detection. Please see [105] for a more detailed discussion.

#### 3.4.1. Evolution of Deep Learning Detectors for Real-Time Performance

R-CNN (Regions with CNN features) is the first deep-learning-based approach to object detection [106]. It extracts candidate proposals by selective search [107] and employs a CNN to extract features from each proposal. The final classification is performed by an SVM classifier. Fast R-CNN [108] extracts features from the entire image and omits redundant computations caused by selective search. Faster R-CNN proposes a Region Proposal Network (RPN) that predicts candidates directly from the shared feature maps [87]. With these advancements, it can achieve significant increases in detection speed. The R-CNN family has been widely used in underwater object detection for a long time [54,109].

The R-CNN family frames object detection as a “coarse-to-fine” process. First, candidate proposals are extracted, then each proposal is classified. These methods are two-stage algorithms. In the other series of detectors, the YOLO family outputs the extraction of candidate region proposals and predicts detection results from shared feature maps directly [44,72,110]. By discarding the extraction of proposals, the inference time can be reduced to 50 ms, whereas other two-stage competitive models require more than 200 ms. In moving from two stages to one stage, detectors have gained a qualitative leap in real-time performance for underwater object detection [34].

In addition, real-time detection has undergone a paradigm shift from anchor-based to anchor-free. The speed of anchor-free detectors is substantially improved over one-stage detectors while maintaining superior detection accuracy, as the expensive operations associated with anchor mechanisms have been eliminated [9]. The de facto method for real-time detection is now anchor-free. As can be seen from the preceding discussion, removing the complex anchor module with its high degree of computational complexity significantly speeds up detection.

#### 3.4.2. Lightweight Network Design

In addition to the evolution of detectors, there are other amazing efforts devoted to real-time detection. One of these is lightweight network design, which aims to develop effective low-complexity network architectures.

Convolution with a 1×1 kernel is the first significant step, and can minimize computational complexity by reducing the number of feature channels. It is extensively utilized in GoogLeNet [111], SqueezeNet [112], etc. The literature shows that 1×1 convolution is a powerful tool for reducing the parameters of deep learning models, resulting in higher detection speeds while maintaining detection accuracy and avoiding degradation [111,112].

Depth-wise separable convolution factorizes a standard convolution into a depth-wise convolution and a 1×1 point-wise convolution, saving a significant amount of operations and parameters while only slightly diminishing accuracy [113]. ShuffleNet proposes point-wise group convolution and channel shuffling, which can drastically reduce computational costs [114].

Next, 1×1 convolution and depth-wise separable convolution are extensively employed to construct lightweight modules, such as the receptive module in [41]. Low-rank approximation is the underlying theory behind lightweight network design. This theory can be used to develop lightweight network architectures by employing different effective and efficient strategies, and is unquestionably crucial in achieving real-time detection of underwater objects.

#### 3.4.3. Model Compression

Model compression aims to remove redundant parameters (or neurons) in pre-trained neural networks. Parameter redundancy has been proven in the literature, and serves as the theoretical foundation of model compression [115].

Various techniques have been developed for model compression. Network pruning removes neurons (or parameters) based on their importance, which is measured by the number of times it is not zero on a given dataset, over its lifetime, or based on another measure [116]. Knowledge distillation uses a small student model to replicate the actions of a large teacher model. It is possible to preserve a large model’s superior performance while shrinking the model’s size and resource usage through knowledge distillation [117]. Parameter quantization aims to reorganize the parameters of neural networks with fewer bits. For example, replacing the 32-bit floating parameters with 16-bit ones is the simplest method. The most commonly used technique for quantization is parameter clustering [118].

Model compression is a thriving research area, and in recent years a plethora of compression strategies have emerged in an effort to achieve an acceptable trade-off between processing speed and application accuracy. Model compression has even influenced the design of neural network accelerators, resulting in the achievement of extreme performance improvements [119].

#### 3.4.4. Summary

In this section, we have reviewed the evolution of underwater object detectors, which is a process pursuing real-time performance. Two key techniques dedicated to real-time detection, namely, lightweight network design and model compression, have been discussed. Lightweight network design aims to create elegant lightweight network architectures, while model compression aims to reduce the redundancy in network parameters. Both are important for real-time detection. In order to create a more elegant model, iteratively applying lightweight network design and model compression is recommended, as these approaches are complementary.

## 4. Popular Datasets

In Section 3, we reviewed the related literature on underwater object detection in light of identified research challenges. In this section, we analyze and discuss the most commonly used datasets for underwater object detection.

### 4.1. Fish4Knowledge

The Fish4Knowledge database was the first well-known popular dataset [17]. It was developed as part of a project conducted at the University of Edinburgh to study marine ecosystems. This dataset enables comparisons and analysis of fish patterns between different water areas [120]. An underwater dataset for target detection against complex backgrounds is one of the ground truth datasets in the Fish4Knowledge. Figure 22 shows example images from the Fish4Knowledge dataset.

**Annotation:** the Fish4Knowledge dataset contains 23 species of fish, and has a total of 27,370 images along with corresponding mask images. All images are annotated manually by marine biologists. Images of different species of fish are stored in different folders. Annotations of bounding boxes for object detection task need to be generated from their mask images.

However, the Fish4Knowledge dataset is not well-suited for fish detection, and researchers need to generate annotations from mask images and create custom training/test sets by themselves in order to meet their special research requirements. In addition, the Fish4Knowledge datavase has a serious class imbalance problem, with certain fish categories dominating their classes.

### 4.2. LifeCLEF 2014

The LifeCLEF2014 dataset was built based on the Fish4Knowledge project, and aims to identify marine organisms in visual data for ecological surveillance and biodiversity monitoring [121]. There are four separate subtasks for fish identification and species recognition in LifeCLEF2014.

For the video-based fish identification subtask, four videos with 21,106 annotations corresponding to 9852 individual fish were fully labeled to form the training dataset, with the aim being to identify moving objects in videos. For the image-based fish identification subtask, the training dataset contains 957 videos with 112,078 labeled fish, with the aim being to detect fish instances in video frames. In the image-based fish identification and species recognition subtask, the training dataset includes 285 videos with 19,868 fishes and the species are annotated, with the aim being to identify species of fish detected in video frames. Image-based fish species recognition is the last challenge, for which the training dataset contains 19,868 annotated fish images and the aim is to identify fish species using only still images containing one fish instance. Matching test datasets are supplied for each subtask.

**Annotation:** the ground truth annotations of the first three subtasks are provided as XML files which contain species, bounding box coordinates, and contour coordinates, as illustrated in Figure 23.

LifeCLEF2014 was significantly extended over the next few years, allowing it to be used with various species of marine organisms as well as in different application tasks. For more information, please refer to [122,123,124]

### 4.3. URPC

The National Natural Science Foundation of China’s Underwater Robot Professional Contest (URPC) offers a well-known dataset for underwater robot object grabbing [55]. Beginning in 2017, the URPC organizer has published a new dataset each year. The images in the URPC databases depict marine organisms such as holothurians, echinus, scallops, and starfish on an open-sea farm. The URPC datasets are widely used in the community of marine science and engineers working on underwater object detection. Figure 24 provides examples of images from the URPC dataset. Table 3 contains information on differnt URPC datasets over the years. Here, ‘Class’ means the categories of objects that need to be detected, while ‘Train’ and ‘Test’ mean the number of images in the training/test sets, respectively.

**Annotation:** the annotations of URPC dataset are provided in XML files, with one file for each image that contains the label and coordinates of the bounding box for each object in the image.

The URPC dataset does not provide annotation files for testing, and cannot be downloaded after the contest ends. Researchers must split the original training data to create a custom training set and test set. This custom split makes the comparison with state-of-the-art methods impossible. In this context, the URPC datasets are unsuitable for use as benchmark datasets.

### 4.4. UDD and DUO

To overcome the shortcomings of the URPC dataset, Dalian University of Technology (DUT) has presented a new dataset (UDD) comprising 2227 photos in three categories (holothurian, echinus, and scallop) to improve the object grasping capabilities of underwater robots for open-sea farming [9]. This dataset comprises 1827 training images and 400 testing images with a maximum resolution of 3840 × 2160 pixels. The UDD is a high-quality and small-scale dataset; hence, it is not a good choice as a benchmark for deep learning-based underwater object detection.

The DUT further gathers and re-annotates the URPC datasets as well as the UDD dataset, yielding 7782 images. Finally, [6] constructed a new Detecting Underwater Objects (DUO) dataset comprising a diversity of underwater scenes and more reasonable annotations. In the DUO dataset, 6671 photos are used for training and 1111 for testing.

**Annotation:** the annotations of the UDD and DUO datasets are prepared in MS COCO format and provided in a JSON file containing the category label, bounding box, segmentation, and other information.

However, the UDD and DUO datasets, similar to the URPC dataset, were generated for the task of grabbing marine organisms using robots. These datasets only have three/four class objects, respectively, and are therefore unsuitable for generic underwater object detection tasks.

### 4.5. Summary

In this section, we have reviewed several popular datasets for underwater marine object detection. At present, there is no universally accepted benchmark dataset, as each of the aforementioned datasets has its own deficiencies. Several of them, such as Fish4Knowledge, are poorly structured, and researchers need to build their own training and testing sets by custom split, making comparison with state-of-the-art methods infeasible [99]. The others are small in scale, which is undesirable for deep learning. These problems have impeded the progress of underwater object detection research. In fact, the lack of annotated data is a major challenge in underwater object detection, prompting the need for further investigation into the construction of a universally accepted benchmark dataset.

## 5. Discussions and Analysis

In this section, we primarily compare our work with previous reviews and discuss which directions in underwater object detection research are important in the future.

### 5.1. Comparison with Previous Reviews

Notable surveys on underwater object detection are summarized in Table 4. The first survey is focused on monitoring of underwater ecosystems [125]. In ecosystem protection research, massive images are frequently collected, prompting a need for automatic object detection and classification. In [125], related works that employ deep learning for underwater imagery analysis are systematically described and categorized according to the category of objects, including fish, plankton, coral, and sea grass. However, a taxonomy based on categorization of objects is not suitable, making high-level understanding of this research topic nearly impossible. On the other hand, the challenges around underwater object detection were barely mentioned in [125].

The second survey reviewed underwater target recognition methods based on deep learning [126]. In [126], the focus was on dangerous underwater target recognition, with methods divided into three categories, namely, target recognition based on shape features, unsupervised recognition techniques, and deep learning methods based on CNNs. The pros and cons of various methods, as well as the challenges in few-shot target recognition and target recognition under environmental interference, were discussed in [126]. Finally, different algorithms were compared on the UDD dataset. The authors concluded that the poor generalization performance of algorithms is always a major problem in this field. However, although a variety of topics in underwater object detection were covered in [126], there was no mention of marine organisms, and a reasonable taxonomy was lacking. The authors briefly identified two challenges in underwater object detection; however, they did not provide a comprehensive analysis.

In another survey of underwater object detection [13], several deep learning approaches and conventional object detection methods that can be deployed on autonomous underwater vehicles were reviewed and a comparison between these methods was carried out. Unfortunately, this survey used a coarse taxonomy, and different research works were only elaborated in the framework of conventional methods and deep learning methods. Moreover, the outstanding challenges in underwater object detection were not presented, and the pros and cons of different methods were not discussed.

The main differences between our paper and existing surveys are highlighted below:**Systematic analysis of research challenges in underwater object detection:** this paper extensively reviews the related literature on underwater object detection. The main challenges around object detection in underwater environments are identified and analyzed systematically. Four challenges, namely, image quality degradation, small object detection, poor generalization, and real-time detection, are discussed at length.**Comprehensive review in light of identified challenges:** an excellent literature review should provide information on the research topic, the challenges researchers are confronted with, and the different solutions that have been proposed. An in-depth exploration of crucial technologies and state-of-the-art methods under the framework of four research challenges makes this survey paper valuable and useful in understanding of underwater marine object detection.**Deeper insight into futuristic trends:** a discussion of how the research topic in question is expected to evolve is an important part of any survey. Based on the state-of-the-art methods in underwater object detection and consideration of next-generation artificial intelligence, this paper provides deeper insight into future trends in underwater object detection.

### 5.2. Future Trends in Underwater Object Detection

Based on the aforementioned discussion, we believe that future directions in underwater marine object detection must include the following aspects:

**(1) A well-constructed benchmark dataset.** The success of deep-learning-based underwater object detection is dependent on the use of a well-constructed benchmark dataset. For any research topic, a good benchmark dataset should be large-scale, diversified, class-balanced, and well-constructed [12,127,128]. The Fish4Knowledge is a large-scale dataset that includes 23 different types of fish species. However, the number of images in each category varies greatly, resulting in a serious class imbalance problem. Although the LifeCLEf datasets are relatively good, they only include fish and ignore other important marine organisms. To the best of our knowledge, no widely accepted benchmark is available at present. Creating a large-scale and well-constructed benchmark dataset is an important topic for future research.

**(2) Inter-class similarities in underwater object detection.** In order to avoid being harmed by predators and improve their survival ability, many marine organisms are very good at camouflage. This presents a challenge in terms of high inter-class similarity, as marine organisms have great similarity with the background or other species. This phenomenon makes it difficult even for humans to recognize marine organisms from their environment, and poses a great challenge for machine learning algorithms. There is no doubt that accounting for subtle differences between objects and their environments can improve detection performance significantly. While it is known that deep learning is relatively robust against inter-class similarities, research works in the area of inter-class similarity are rare [24]. Addressing this problem is an important challenge in moving underwater marine object detection forward.

**(3) Underwater object detection based on transformers.** One of the biggest surprises in artificial intelligence research in recent years is the development of transformers. As a promising fundamental infrastructure, transformers have dominated research on object detection during the 2020s thanks to competitive performance and tremendous potential [129]. However, the most famous vanilla transformers are used with large models with high computational complexity, and require very long training times to converge. Due to these intrinsic limitations, transformers are much slower than traditional convolution with the same FLOPs (at least 3× slower) [130]. In addition, transformer suffer from a relatively low performance on small object detection tasks due to their non-local sensitivity [131]. Considering the constraints in underwater environments, an efficient transformer designed specifically for small objects could be of great significance for underwater marine object detection.

**(4) Multi-modal data fusion.** Most research studies on underwater object detection rely solely on optical or acoustic sensors. However, it is well-known that each of these sensors has its own distinct characteristics and limitations. Optical imaging provides high imaging resolution and rich information, as well as being more intuitive for humans. However, the sensing range and image quality are severely influenced by absorption and scattering effects. Sonar imaging based on acoustic waves, on the other hand, has the advantages of low loss and long propagation distance. The active [132,133] and passive [134] acoustic approaches continue to play an important role in underwater exploration. Hence, complementary employment of these two kinds of methods represents a way to greatly improve object detection performance. Multi-modal data fusion is highly desirable in underwater exploration.

**(5) Multi-task learning.** Rather than extending existing models to new tasks, today’s AI models are typically trained to perform only one task. However, the human brain does not work in this manner. A variety of capabilities that can be accessed as needed and combined to perform new and/or more complex tasks is highly desirable [135]. Due to the challenges of image quality degradation, new image enhancement and/or restoration techniques are always beneficial. Therefore, engaging both underwater image enhancement and object detection in a well-designed multi-task learning framework is a promising direction. Multi-task learning can improve data efficiency as well, as this paradigm does not require learning every task from scratch.

**(6) Sparse model with efficient computing.** It is widely accepted that the success of deep learning is due to the unprecedented availability of big data and concurrent advancements in computing power [136]. In the traditional dense model, the entire neural network is activated to complete a task regardless of how simple or complex it is [137], which is inefficient and goes against the way the human brain works. Creating a sparse model in which only small segments of the network are activated as needed can, in theory, result in a higher capability for different tasks, and is more energy efficient as well. Sparse models with efficient computing can undoubtedly make multi-task learning more feasible, and represent a key technological infrastructure for the next generation of ocean exploration techniques.

## 6. Conclusions

Deep learning-based underwater object detection has received significant attention in the community of marine science and engineering researchers thanks to its superior performance. It has tremendous potential to support a wide range of marine activities. This paper presents a comprehensive and critical review of deep learning-based underwater object detection techniques. Four research challenges in the underwater environment are identified in this paper, namely, image quality degradation, small object detection, poor generalization, and real-time detection. In light of these identified challenges, a comprehensive analysis is presented to provide a thorough understanding of the subject matter. Finally, popular datasets and future directions in underwater object detection are discussed. We hope that readers find this survey informative and useful in helping them to understand the recent advances in underwater object detection, and that it can guide future research in this exciting area.

## Figures and Tables

**Figure 1 sensors-23-01990-f001:**
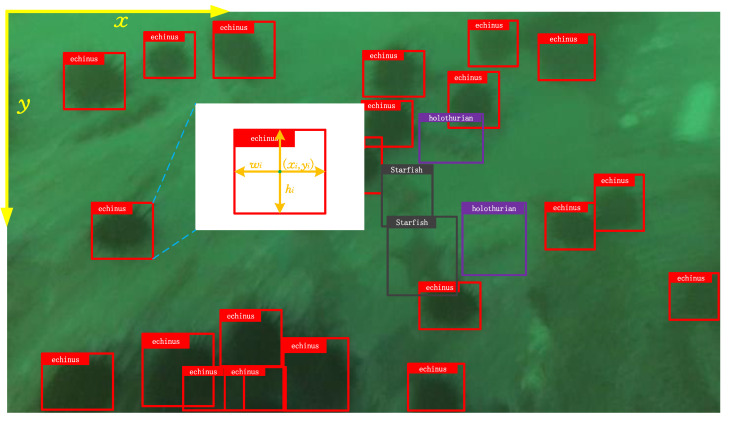
Illustration of vision-based underwater object detection. The detection result is presented as a bounding box with a label, where $\left(x_{i},y_{i}\right)$ denotes the *i*th object’s center coordinates and $\left(w_{i},h_{i}\right)$ denotes the box’s width and height. Here, (x,y) are the axis frames, with the origin in the image’s upper left corner (image from DUO dataset [6]).

**Figure 2 sensors-23-01990-f002:**
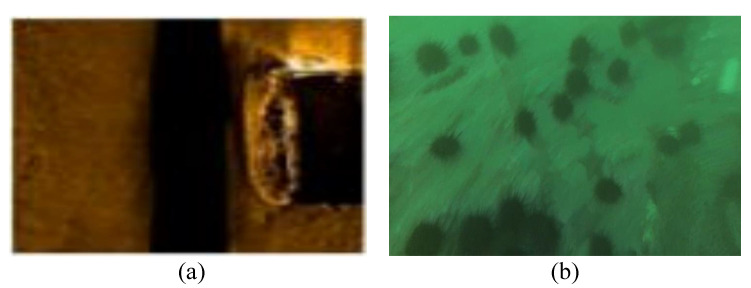
Illustration of (**a**) sonar image and (**b**) camera image (images from [11] and DUO dataset [6]), respectively.

**Figure 3 sensors-23-01990-f003:**
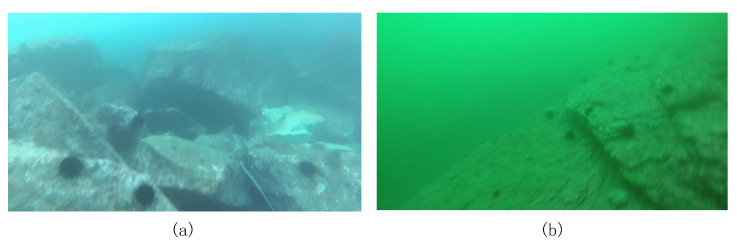
Most underwater images have (**a**) a bluish or (**b**) an aqua tone, which is due to the selective absorption of light in open water (images from DUO dataset [6]).

**Figure 4 sensors-23-01990-f004:**
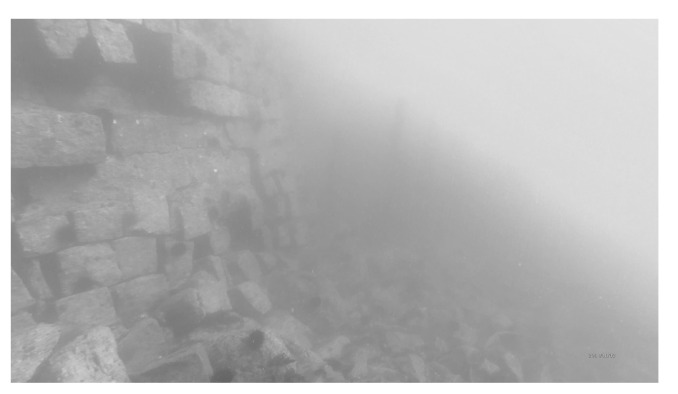
Illustration of blurry image caused by scattering (image from DUO dataset [6]).

**Figure 5 sensors-23-01990-f005:**
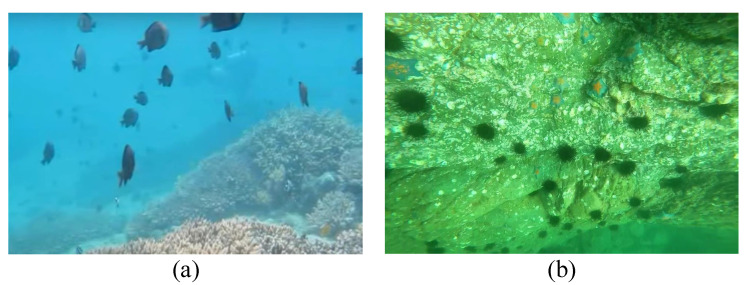
(**a**) Fish school and (**b**) benthic organisms (images from Fish4knowledge dataset [17] and DUO dataset [6]), respectively.

**Figure 6 sensors-23-01990-f006:**
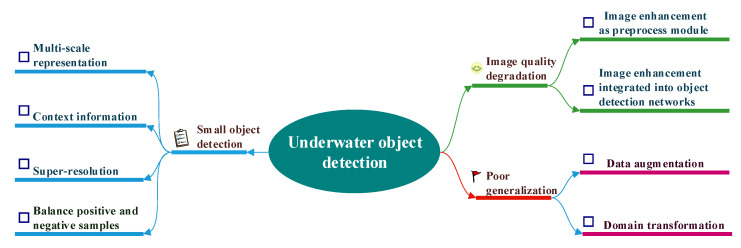
Framework of underwater object detection according to the main challenges identified in underwater environments.

**Figure 7 sensors-23-01990-f007:**
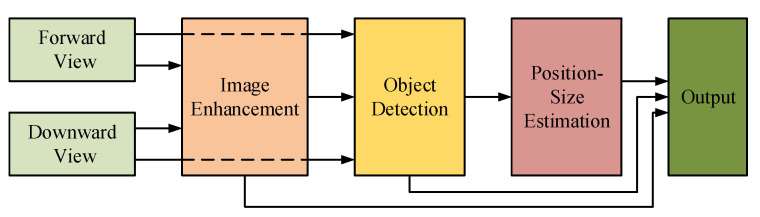
Workflow of the underwater sensing system proposed in [26].

**Figure 8 sensors-23-01990-f008:**
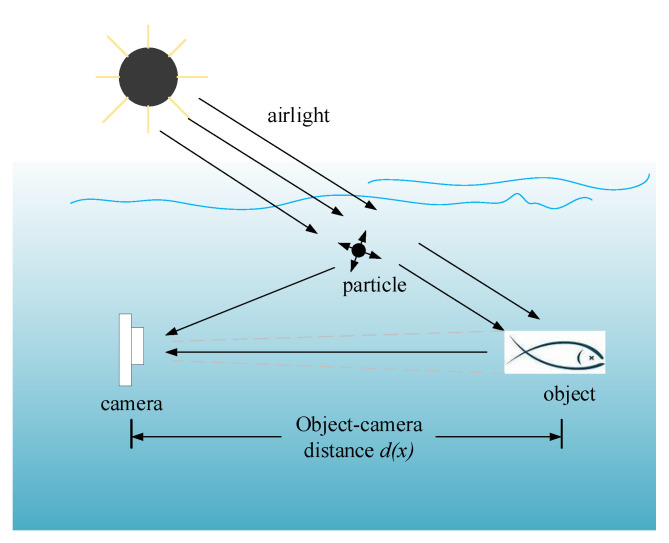
Simplified image formulation model in underwater environment, as proposed in [35].

**Figure 9 sensors-23-01990-f009:**
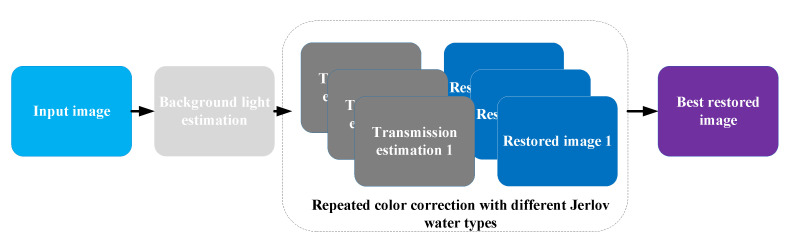
The color restoration and transmission estimation method proposed in [36].

**Figure 10 sensors-23-01990-f010:**
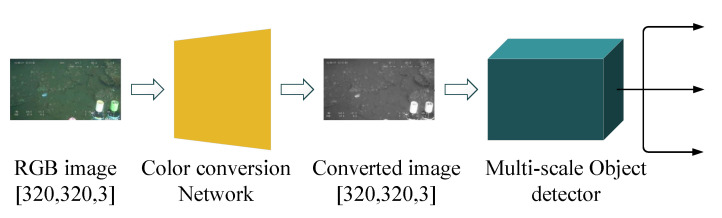
Joint learning of color conversion and object detection in underwater images (figure from [41]).

**Figure 11 sensors-23-01990-f011:**
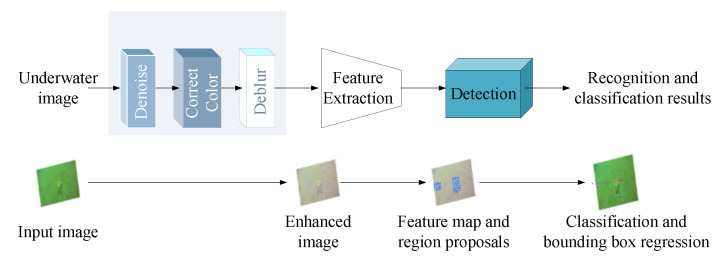
Marine organism detection framework with joint optimization of both image enhancement and object detection (figure from [45]).

**Figure 12 sensors-23-01990-f012:**
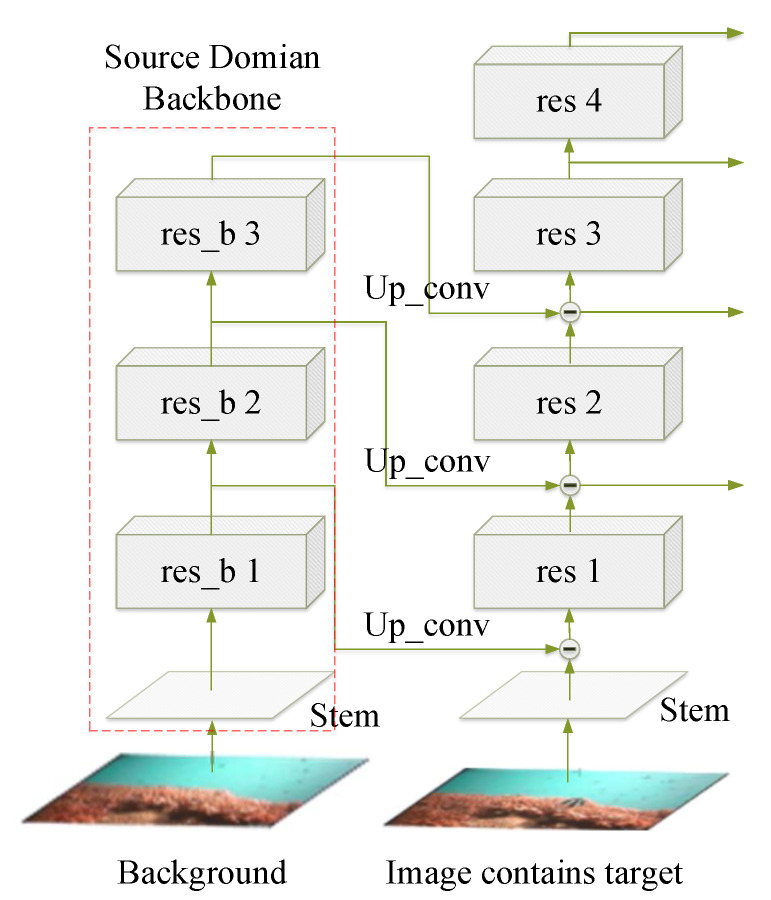
CBresnet backbone network; the interference is encoded and eliminated by the Source Domain Backbone (figure from [46]).

**Figure 13 sensors-23-01990-f013:**
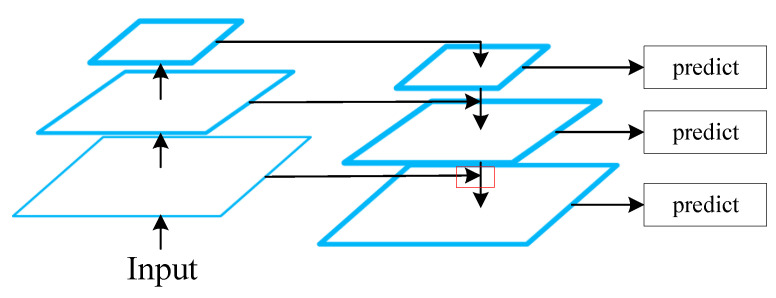
Structure of Feature Pyramid Network (figure from [47]).

**Figure 14 sensors-23-01990-f014:**
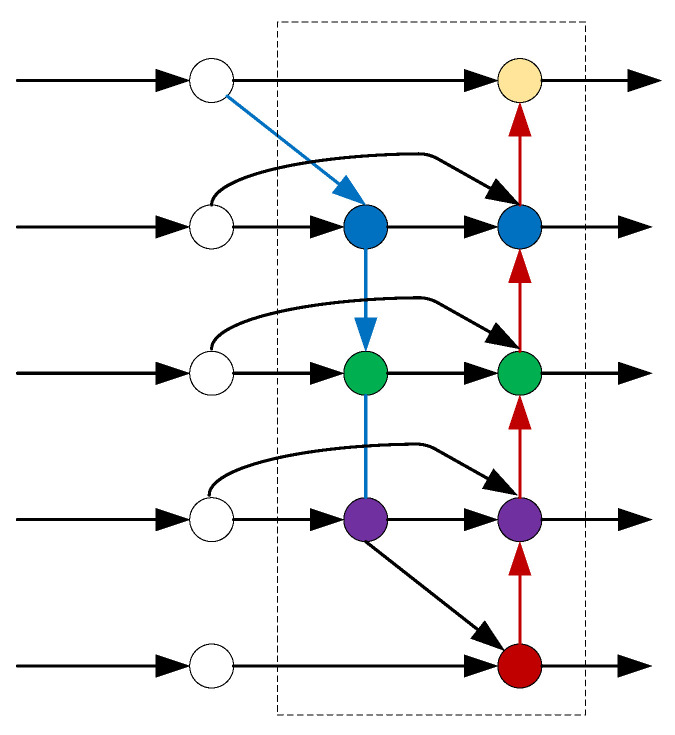
Structure of Bidirectional Feature Pyramid Network (figure from [53]).

**Figure 15 sensors-23-01990-f015:**
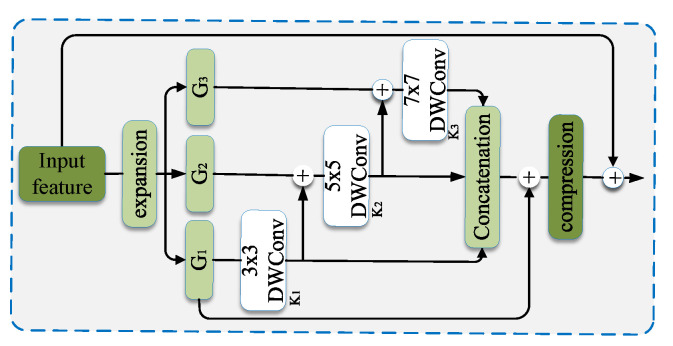
Multi-scale Contextual Features Fusion (MFF) block (figure from [9]).

**Figure 16 sensors-23-01990-f016:**
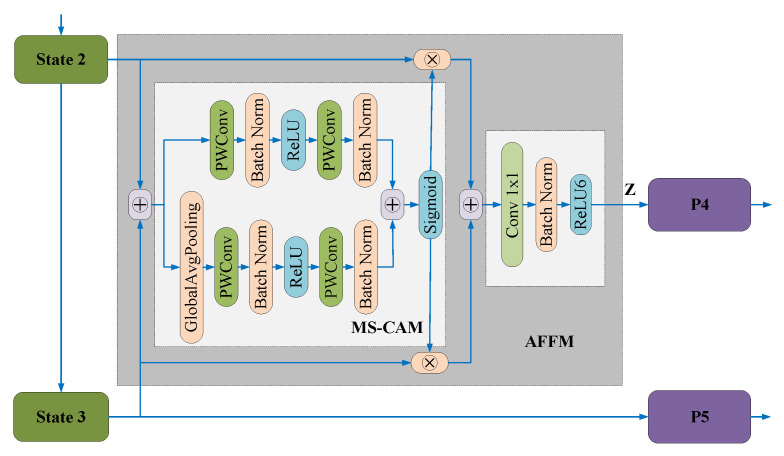
Multi-scale Attentional Feature Fusion Module (figure from [60]).

**Figure 17 sensors-23-01990-f017:**
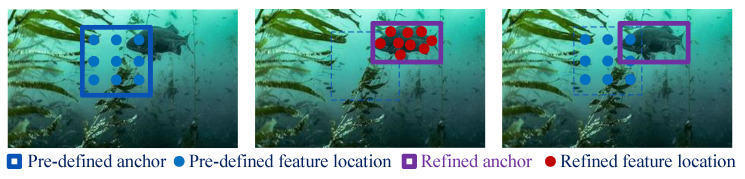
Illustration of Anchor Refinement and Feature Location Refinement mechanisms proposed in [65].

**Figure 18 sensors-23-01990-f018:**
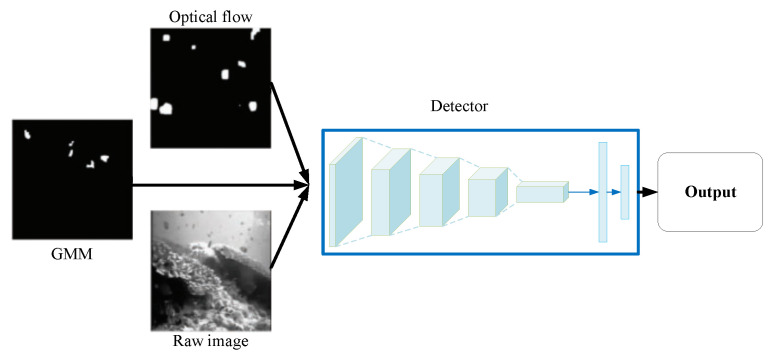
Combining motion information for small underwater object detection (figure from [66]).

**Figure 19 sensors-23-01990-f019:**
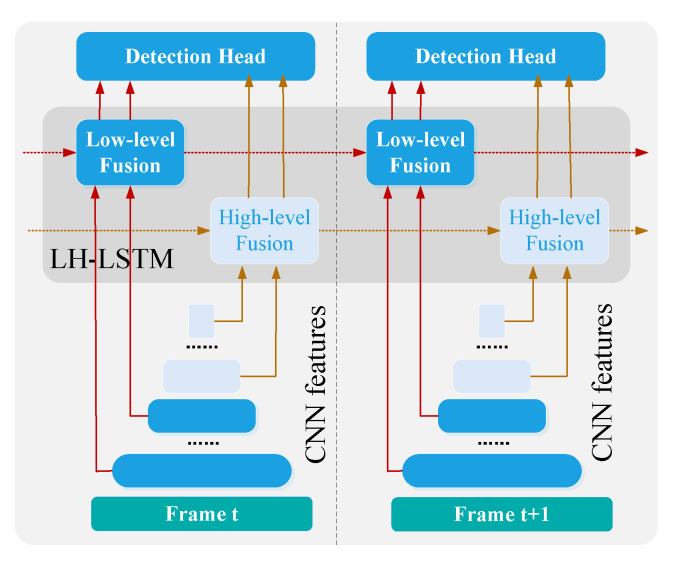
Schematic illustration of TSSD proposed in [70].

**Figure 20 sensors-23-01990-f020:**
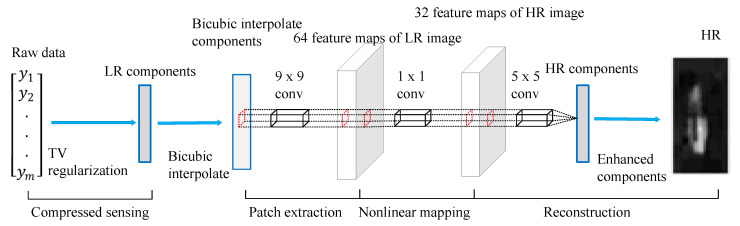
Reconstruction of Underwater Single-Pixel Imaging based on CS-SRCNN (figure from [76]).

**Figure 21 sensors-23-01990-f021:**
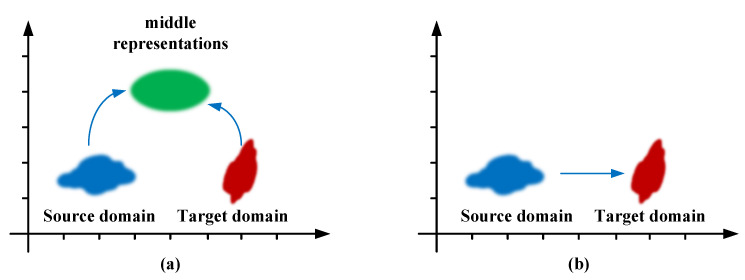
Two kinds of domain transformation for generalization in underwater object detection, where (**a**) transforms distinct domains to middle representations and (**b**) transform source domain to target domain.

**Figure 22 sensors-23-01990-f022:**
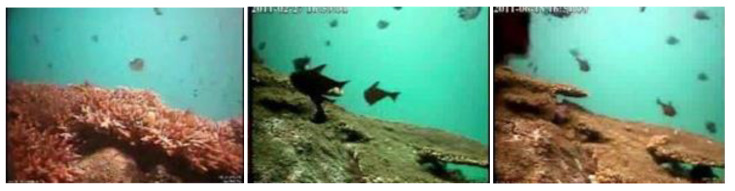
Example images from the Fish4Knowledge dataset (images from [17]).

**Figure 23 sensors-23-01990-f023:**
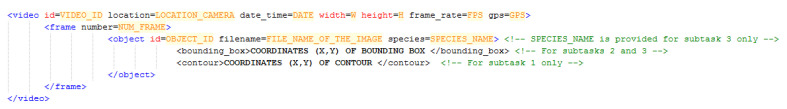
Ground truth annotations of LifeCLEF2014 dataset (image from ImageCLEF website).

**Figure 24 sensors-23-01990-f024:**
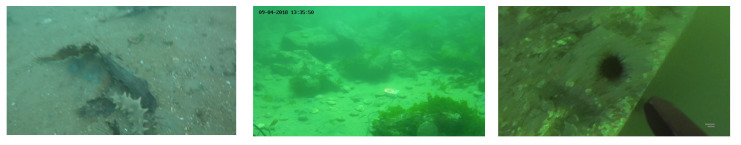
Example images from the URPC dataset.

**Table 1 sensors-23-01990-t001:** Comparison of different methods on custom dataset collected in [41]. The numbers in bold indicate the best.

Metric	No Conversion	As Preprocessing	Proposed
**mAP**	0.878	0.844	**0.896**
**GFLOPs**	**4.88**	5.02	5.06
**Epochs**	107	81	**76**

**Table 2 sensors-23-01990-t002:** Comparison of different feature fusion networks on SeaLife 2017 dataset [46]. The numbers in bold indicate the best.

Methods	AP	${\mathrm{AP}}_{50}$	${\mathrm{AP}}_{75}$
**FPN**	0.727	0.921	**0.875**
**PANet**	0.733	0.917	0.866
**EPANet**	**0.741**	**0.924**	0.872

**Table 3 sensors-23-01990-t003:** Information on URPC datasets over years.

Dataset	Train	Test	Class	Year
URPC2017	17,655	985	3	2017
URPC2018	2901	800	4	2018
URPC2019	4757	1029	4	2019
URPC2020_ZJ	5543	2000	4	2020
URPC2020_DL	6575	2400	4	2020

**Table 4 sensors-23-01990-t004:** Summary of related surveys on underwater object detection.

No.	Survey Title	Content	Deficiency	Year	References
1	Deep Learning in Underwater Marine Object Detection: A Survey	Summarized deep learning object detection and classification methods with respect to the categories of objects, including fish, plankton, coral, and sea grass.	Unsuitable taxonomy; the main challenges of underwater object detection were not discussed.	2017	[125]
2	Underwater Target Recognition Methods Based on the Framework of Deep Learning: A Survey	Described the application of deep learning in underwater dangerous target and man-made object recognition. Discussed two main challenges around few-shot learning and environmental interference.	Does not contain marine organisms; used unsuitable taxonomy; analysis of challenges not comprehensive.	2020	[126]
3	Robust Underwater Object Detection with Autonomous Underwater Vehicles: A Comprehensive Study	Briefly discussed conventional methods and deep learning methods for underwater object detection with AUVs.	Challenges in underwater object detection were not discussed; taxonomy simple and crude; similarities and differences between two categories of methods not described.	2020	[13]

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
