# Peer review of "Research Challenges, Recent Advances, and Popular Datasets in Deep Learning-Based Underwater Marine Object Detection: A Review"

_sensors, 2023, doi:10.3390/s23041990_

Round 1

Reviewer 1 Report

In this manuscript the authors review and critically discuss the recent literature about the application of deep learning methods to underwater object detection. They list several challenges posed by the underwater environment, which makes deep learning more difficult to apply in this scenario compared to classical computer vision settings. Finally, they mention some popular datasets that are used by the research community in this field, and propose a series of promising future research directions that could be explored to make further progress.

I enjoyed reading this manuscript, which is nicely written and balanced. It also contains several explanatory figures and it covers most of the relevant literature. At the same time, I have a few key concerns that should be addressed to improve the overall quality and make the paper stronger:

1) In my view, a critical issue with the current manuscript is that “object detection” is implicitly assumed to be equivalent to “visual object detection”. However, object detection can be performed in many other ways, and from the title and abstract I was expecting a more comprehensive review of the literature that also covered, for example, acoustic monitoring approaches. In fact, in underwater scenarios it is often more convenient to perform object detection by relying on sound reflections rather than optical information (which can be degraded in night-time and in turbid waters). It would be a great plus to also cover some representative active (e.g., https://doi.org/10.1109/TMC.2020.3044397, https://doi.org/10.3390/s20102945) and passive (e.g., https://doi.org/10.26748/KSOE.2020.017) acoustic monitoring approaches that exploited deep learning for underwater object detection. If the authors prefer to focus their paper on visual object detection, they could just mention these alternative approaches in the section named “Futuristic Trends of Underwater Object Detection” and amend the abstract to clarify that the paper will be focused on visual input.

2) An additional challenge that the authors might consider discussing is that of energy-consumption, since deep learning models often requires high-performance computing hardware that is not easy to deploy in underwater monitoring platforms. What are the current approaches to tackle this issue, and how well such approaches perform in underwater scenarios?

3) The lack of annotated data is another big challenge in underwater signal processing. It would be good to stress this issue, and to better highlight the kind of annotations available in each of the datasets discussed in Section 4.

4) When discussing Image Enhancement, the authors might also mention advanced color restoration methods that have proven particularly useful in underwater settings (https://doi.org/10.1109/TPAMI.2020.2977624).

5) When discussing the challenge of Generalization, the authors might also mention the use of transfer learning approaches, which have proven extremely effective in underwater object detection (https://doi.org/10.1109/CVPRW.2018.00187).

Minor issues:

- Line 49: “However, due to interferences from the underwater environment, such as complex background structures, characteristics of marine objects, and limitations imposed by exploration equipment, underwater object detection has become an extremely challenging task.” -> I would say that underwater object detection has always been challenging. So probably it should be rephrased in “…underwater object detection is an extremely challenging task”.

- Line 150: “there is a significant imbalance between positive and negative examples in small object detection, which is not conducive to model training because background examples will dominate training loss and gradient updates will be too deviated to learn features of foreground positive examples” -> this sentence is a bit cryptic: are the authors arguing that the background objects are usually smaller (and thus more challenging to detect)?

- What are the a and b axes representing in Fig. 6 and Fig. 22? This should be described at least in the figure caption.

- Fig. 14 should be better explained in the text and in the caption itself, possibly also adding some labels in the graph to more clearly communicate the kind of information processed at different nodes / layers.

- Line 365: missing reference and figure.

- Line 399: signoid -> sigmoid

Author Response

Response to Reviewer 1 Comments

Point 1: In my view, a critical issue with the current manuscript is that “object detection” is implicitly assumed to be equivalent to “visual object detection”. However, object detection can be performed in many other ways, and from the title and abstract I was expecting a more comprehensive review of the literature that also covered, for example, acoustic monitoring approaches. In fact, in underwater scenarios it is often more convenient to perform object detection by relying on sound reflections rather than optical information (which can be degraded in night-time and in turbid waters). It would be a great plus to also cover some representative active (e.g., https://doi.org/10.1109/TMC.2020.3044397, https://doi.org/10.3390/s20102945) and passive (e.g., https://doi.org/10.26748/KSOE.2020.017) acoustic monitoring approaches that exploited deep learning for underwater object detection. If the authors prefer to focus their paper on visual object detection, they could just mention these alternative approaches in the section named “Futuristic Trends of Underwater Object Detection” and amend the abstract to clarify that the paper will be focused on visual input.

Response 1: In this paper, we prefer to focus on visual object detection. So, we have amended the abstract to clarify that the paper will be focused on visual input. We also explain the reason that why we focus on visual input in the Section #2.1. Preliminaries. Optical cameras can capture a lot of semantic information, which is beneficial for the recognition of target objects. Another major benefit of optical cameras is their inexpensive price, which makes them a promising sensor for underwater exploration. We also have mentioned the active and passive acoustic monitoring approaches in Section # 5.2. Futuristic Trends of Underwater Object Detection: 4) Multi-modal data fusion. We claim that complementary employment of optical and acoustic approaches will be highly desired in underwater exploration.

Point 2: An additional challenge that the authors might consider discussing is that of energy-consumption, since deep learning models often requires high-performance computing hardware that is not easy to deploy in underwater monitoring platforms. What are the current approaches to tackle this issue, and how well such approaches perform in underwater scenarios?

Response 2: We agree energy consumption is an additional challenge that should be considered. We named the problem of energy consumption as "challenge of real-time detection." Huge energy-consumption makes real-time detection impossible in the commonly used marine robots, such as such as Remotely Operated Underwater Vehicles (ROVs) and Autonomous Underwater Vehicles (AUVs), which only have limited computing power. We have put this challenge in a separate section in this paper.

Point 3:  The lack of annotated data is another big challenge in underwater signal processing. It would be good to stress this issue, and to better highlight the kind of annotations available in each of the datasets discussed in Section 4.

Response 3: We agree that the lack of annotated data is another big challenge in underwater object detection. Hence, we have claimed in Section # 5.2. Futuristic Trends of Underwater Object Detection: 1) A well-constructed benchmark dataset that -- "To our knowledge, no widely accepted benchmark is available. Creating a large-scale and well-constructed benchmark dataset will be an important topic for future research."  As you suggested, we have also highlighted the kinds of annotations available in each of the datasets discussed in Section 4.

Point 4:  When discussing Image Enhancement, the authors might also mention advanced color restoration methods that have proven particularly useful in underwater settings (https://doi.org/10.1109/TPAMI.2020.2977624).

Response 4: The advanced color restoration methods (https://doi.org/10.1109/TPAMI.2020.2977624) have been mentioned in Section #3.1.1 Image Enhancement as Preprocessing Module.

Point 5: When discussing the challenge of Generalization, the authors might also mention the use of transfer learning approaches, which have proven extremely effective in underwater object detection (https://doi.org/10.1109/CVPRW.2018.00187).

Response 5: Transfer learning approaches have been mentioned in Section #3.3.2 Domain Transformation. Transfer learning alleviates the challenge of generalization by applying the knowledge gained from source domain to target domain.

Minor issues:

- Line 49: “However, due to interferences from the underwater environment, such as complex background structures, characteristics of marine objects, and limitations imposed by exploration equipment, underwater object detection has become an extremely challenging task.” -> I would say that underwater object detection has always been challenging. So probably it should be rephrased in “…underwater object detection is an extremely challenging task”.

Response: we have rephrased this sentence.

- Line 150: “there is a significant imbalance between positive and negative examples in small object detection, which is not conducive to model training because background examples will dominate training loss and gradient updates will be too deviated to learn features of foreground positive examples” -> this sentence is a bit cryptic: are the authors arguing that the background objects are usually smaller (and thus more challenging to detect)?

Response: No. As we know, a machine learning model always tends to predict a sample as the dominant class in a class imbalance scenario. For example, we have 90% negative samples and 10% positive samples in our classification dataset. A machine learning model only needs to predict all samples as negative to obtain a high precision (90%). But this model is useless. In the gradient-descent-based methods, class-imbalance problem will lead to the fact that most of the loss is contributed by negative samples. So do gradients. This phenomenon will lead machine learning models to ignore the feature learning of positive samples and predict all samples as negative. This is not desired.

- What are the a and b axes representing in Fig. 6 and Fig. 22? This should be described at least in the figure caption.

Response: Lab color space is a color space defined by the International Commission on Illumination (abbreviated CIE) in 1976. It expresses color as three values: L for perceptual lightness and a and b for the four unique colors of human vision: red, green, blue and yellow. The lightness value, L, defines black at 0 and white at 100. The a axis is relative to the green–red opponent colors, with negative values toward green and positive values toward red. The b axis represents the blue–yellow opponents, with negative numbers toward blue and positive toward yellow. But we have removed Fig. 6 and Fig. 22 now.

- Fig. 14 should be better explained in the text and in the caption itself, possibly also adding some labels in the graph to more clearly communicate the kind of information processed at different nodes / layers.

Response: We have added explanation for Fig.14 in the text and in the caption itself.

- Line 365: missing reference and figure.

Response: We have rephrased this sentence.

- Line 399: signoid -> sigmoid

Response: We have corrected this word.

Reviewer 2 Report

The paper is well-written and fits well the journal's scope. But before publishing it, some concerns need to be clarified. 

1. As a review paper, how did you select the papers for the review? Which criteria did you use? Did you select them by searching in any global database, e.g. Pubmed, Scopus, etc...

2. Some figures come from other state-of-art papers do you have their copyright?

3. As a review paper, I would expect to find a summarized table regarding the precision and recall of all consulted state-of-art papers.

4. some figures are not indicated in the text, e.g. line 365 with  Figure with  ??

Author Response

Response to Reviewer 2 Comments

Point 1: As a review paper, how did you select the papers for the review? Which criteria did you use? Did you select them by searching in any global database, e.g. Pubmed, Scopus, etc...

Response 1: We first searched and collected papers on Google Scholar and Web of Science database using "deep learning, underwater object detection" as key words. We also collected papers from references when we read the papers that we had collected before. We select the papers based on whether they are relevant to our topic, and then assess these papers based on their novelty, authority (or credibility), whether they were published in recent years, etc.

Point 2: Some figures come from other state-of-art papers do you have their copyright?

Response 2: Yes. Most figures are open access. And we have removed several figures that were not copyrighted

Point 3: As a review paper, I would expect to find a summarized table regarding the precision and recall of all consulted state-of-art papers.

Response 3: Yes. A comparison between state-of-the-art methods on a benchmark dataset should be included in review paper. However, to our knowledge, no widely accepted benchmark is available. On the other hand, many state-of-art papers do not open source their code, making the comparison impossiable.

Point 4: Some figures are not indicated in the text, e.g. line 365 with  Figure with  ??

Response 4: We have corrected this mistake.

Reviewer 3 Report

Underwater marine object detection is one of the most fundamental techniques in the community of marine science and engineering. It has tremendous potential for exploring the ocean. This manuscript reviews the research challenges, recent advances and benchmark datasets in deep-learning-based underwater marine object detection. The manuscript is well organized and written. The content of the manuscript is comprehensive. I think it will be a good reference for researchers in the field of marine object detection as well as generic object detection. 

Author Response

Response to Reviewer 3 Comments

Comments and Suggestions for Authors: Underwater marine object detection is one of the most fundamental techniques in the community of marine science and engineering. It has tremendous potential for exploring the ocean. This manuscript reviews the research challenges, recent advances and benchmark datasets in deep-learning-based underwater marine object detection. The manuscript is well organized and written. The content of the manuscript is comprehensive. I think it will be a good reference for researchers in the field of marine object detection as well as generic object detection.

Response : Thank you for your approval.

Reviewer 4 Report

This article reviews recent techniques in underwater object detection. The authors point out some challenges in this topic. I only have two concerns.

1. line 338: the authors said that "the mAP increased 27.21%". I would like to know which method is the benchmark for comparison.

2. The authors describe the problems and corresponding solutions in terms of detecting small objects, improvements of image quality and better generalization. Can you provide a quick summary for each problem, as you did in section 4? I believe it will give readers a better understanding of these methods.

Author Response

Response to Reviewer 4 Comments

Point 1: line 338: the authors said that "the mAP increased 27.21%". I would like to know which method is the benchmark for comparison.

Response 1: The benchmark method is the original YOLO-v4.

Point 2: The authors describe the problems and corresponding solutions in terms of detecting small objects, improvements of image quality and better generalization. Can you provide a quick summary for each problem, as you did in section 4? I believe it will give readers a better understanding of these methods.

Response 2: Yes. We have provided a summary for each problem now.

Round 2

Reviewer 2 Report

The authors clarified all my comments, the paper is ready for publication.